# Status of Metabolomic Measurement for Insights in Alzheimer’s Disease Progression—What Is Missing?

**DOI:** 10.3390/ijms24054960

**Published:** 2023-03-04

**Authors:** Chunyuan Yin, Amy C. Harms, Thomas Hankemeier, Alida Kindt, Elizabeth C. M. de Lange

**Affiliations:** 1Metabolomics and Analytics Centre, Leiden Academic Centre for Drug Research, Leiden University, 2333 CC Leiden, The Netherlands; 2Division of Systems Pharmacology and Pharmacy, Leiden Academic Centre for Drug Research, Leiden University, 2333 CC Leiden, The Netherlands

**Keywords:** Alzheimer’s disease, metabolomics, lipidomics, biomarkers, pathways, animal, human

## Abstract

Alzheimer’s disease (AD) is an aging-related neurodegenerative disease, leading to the progressive loss of memory and other cognitive functions. As there is still no cure for AD, the growth in the number of susceptible individuals represents a major emerging threat to public health. Currently, the pathogenesis and etiology of AD remain poorly understood, while no efficient treatments are available to slow down the degenerative effects of AD. Metabolomics allows the study of biochemical alterations in pathological processes which may be involved in AD progression and to discover new therapeutic targets. In this review, we summarized and analyzed the results from studies on metabolomics analysis performed in biological samples of AD subjects and AD animal models. Then this information was analyzed by using MetaboAnalyst to find the disturbed pathways among different sample types in human and animal models at different disease stages. We discuss the underlying biochemical mechanisms involved, and the extent to which they could impact the specific hallmarks of AD. Then we identify gaps and challenges and provide recommendations for future metabolomics approaches to better understand AD pathogenesis.

## 1. Introduction

Alzheimer’s disease (AD), the most common form of dementia in aging population, leads to the progressive loss of memory and other cognitive functions [1]. In 1907, Dr. Alois Alzheimer discovered the first patient with senile plaques and NFTs, which represent the major hallmarks of AD which has become a major public health problem due to the increase in the elder population worldwide [2]. AD can be divided into two types: familial AD (5%) and sporadic AD (95%). Familial AD (FAD) has an early onset (<65 years of age) and it is caused by mutations in the genes encoding amyloid precursor protein (APP) and presenilin 1 and 2 (PS1 and PS2) [3]. Age is a major risk factor for AD, but inactivity (lack of exercise), obesity, diabetes, high blood pressure, high cholesterol, and too high alcohol consumption also increase the incidence of AD. Furthermore, it seems that low educational level, social isolation, and cognitive inactivity also contributes to AD [4]. Today, diagnosis of AD is based on several neuropsychological tests, imaging, and biological analyses, which all indicate AD in a later stage. Currently, there is no efficient treatment available, although treatment by the recently approved drug lecanemab seems to delay AD progression [5], and provides some hope.

Currently, we know that brain extracellular amyloid deposits, called neuritic senile or amyloid-β plaques, and fibrillary protein deposits inside neurons, known as neurofibrillary bundles or tau tangles, appear mainly in the frontal and temporal lobes and contribute to AD progression [6]. However, there are still many questions about how AD initiates and how it progresses.

Information on factors in and mechanism of initiation and progression, therefore, is crucial for earlier diagnosis, as well as to find targets to treat AD in a stage-dependent manner. AD is very complex, and a systems biological approach is warranted. Metabolomics allows such an approach as it can be used to measure biochemical alterations underlying pathological processes thus offering great potential for the diagnosis and prognosis of neurodegenerative diseases. This is because a subject’s metabolome reflects alterations in genetic, transcript, and protein profiles as well as influences from the environment [7]. Moreover, since metabolic pathways are largely conserved between species, metabolomics could improve the translation of preclinical research conducted in animal models of AD into humans. Furthermore, the brain has a high lipid content, which indicates that lipidomics may be a highly valuable omics technique as well, to provide novel insights into AD pathogenesis.

In this review, we provide an overview of the current state of application of metabolomics (including lipidomics) research on AD in human and animal models, together with the metabolite information that has been obtained from plasma, brain, and cerebrospinal fluid (CSF) samples in human studies and animal studies. We then analyze this information using MetaboAnalyst to find the disturbed pathways among different sample types in human and animal models of different disease stages. Then we identify gaps and challenges and provide recommendations for future metabolomics approaches to better understand AD pathogenesis.

## 2. Metabolomics

Metabolites are substrates, intermediates, and products of metabolic body processes, which typically are small molecules with a molecular weight of less than ~1.5 kDa [8]. Since low molecular weight metabolites are intermediates or end products of cellular metabolism, metabolomics, or the study of metabolism can be considered one of the core disciplines of systems biology. It can help in improving our understanding of changes in biochemical pathways, revealing crucial information that is closely related to human disease or therapeutic status [9,10,11,12,13].

Metabolomics allows the systematic study of unique metabolomic fingerprints that result from the body functioning in different conditions, such as healthy and diseased. These fingerprints can be viewed as biomarkers of normal biological processes, pathological processes, or pharmacological responses to a therapeutic intervention [14,15,16]. 

Metabolism refers to the biochemical reactions that occur throughout the body within each cell and that provide the body with energy. This energy is needed for vital body processes and the synthesis of new body components [17]. Some of these are mediated by enzymes, with specialized functions in anabolism and catabolism. To that end, the body needs nutrients and energy that come from the diet. Metabolism is affected by many factors such as sex, race, exercise, diet, age, and diseases such as Parkinson’s or Alzheimer’s. The biggest impacts on metabolites are genetics and environment. 

Lipidomics, a subfield of metabolomics, is the study of the lipidome, i.e., all the lipids within cells, organs, or biological systems. Lipids are vital in the biological processes of living organisms. They not only serve as structural components of cell membranes, but also play an important role in the source of chemical energy and cell signaling molecules [18]. Apart from adipose tissue, which is the most lipid-rich organ, the brain is the body’s second lipid-rich organ; 10% to 12% of the fresh weight and more than 50% of the dry weight is composed of lipids [19]. With significant structural diversity, major lipid species in the brain can be categorized as sphingolipids, glycerolipids, glycerophospholipids, fatty acids, cholesterol, and cholesterol ester. Among them, phospholipids account for 50% of total lipid content [20]. Plasma abnormal lipid profiles have been known to be associated with AD for several decades [21,22,23]. Lipidomics profiling of plasma and tissues has the potential to discover biomarkers of aging or AD, which can contribute to understanding the pathological mechanism of AD.

### 2.1. Metabolomics Platforms for Identification of AD Biomarkers

There are a number of metabolomics platforms available, each with its specific advantages and disadvantages, most notably differences in sensitivity, reproducibility, and equipment costs [24]. Due to the diversity of the metabolome and the complexity of biological systems, it is impossible to give a fully comprehensive metabolite profile of a biological sample by using a single analytical platform [25]. Therefore, the choice of analytical platforms will be determined by the nature of the biological specimen to be analyzed, the goal of the analysis, the nature of the compounds under investigation (i.e., polar or apolar, volatile or non-volatile), and the resources of the laboratory [24,26,27]. Numerous metabolomics platforms are commonly used in both targeted and untargeted studies, and include gas chromatography-mass spectrometry (GC-MS), liquid chromatography-mass spectrometry (LC-MS), capillary electrophoresis-mass spectrometry (CE-MS), direct-infusion mass spectrometry (DI-MS), and nuclear magnetic resonance (NMR).

GC-MS is generally considered a versatile platform given its excellent separation power, sensitivity, and reproducibility [28]. This platform often requires a chemical derivatization procedure to create volatile compounds, which means the compounds profiled are limited to those that are volatile or can be made volatile using this complex and time-consuming procedure [29]. In addition, derivatization can improve volatility, thermal stability, sensitivity, chromatographic selectivity, and peak shapes [30]. GC-MS uses electron ionization (EI) or chemical ionization (CI) to analyze volatile metabolites. EI-MS is a hard ionization technique that does not suffer from ion suppression, which means it can generate quantitative data and extensive and predictable fragmentation for the structural characterization of metabolites [31]. As EI mass spectra are consistent across instruments and laboratories, sample identification in GC-MS is based on the use of EI mass spectral libraries by matching mass spectral fragment ion patterns, which can be seen as a compound-specific “fingerprint” [32,33].

LC-MS, which is also known as high-performance LC-MS (HPLC-MS) or ultra-HPLC (UHPLC or UPLC), is predominantly used in metabolomics and can provide analysis of thermally non-volatile, unstable, high- or low-molecular weight compounds with wide polarity range. LC-MS does not need a derivatization step, which makes sample preparation simpler and more amenable to high throughput analysis [25]. The columns used in liquid chromatography separate metabolites based on the physical properties of the molecules. Two classes of stationary phases commonly used in metabolomics analysis are hydrophilic interaction liquid chromatography (HILIC) and reversed-phase (RP). HILIC is good at analyzing highly hydrophilic and ionic compounds and therefore suitable for profiling polar metabolites, whereas RP with C8 or C18 columns is widely used in providing good separation of non-polar or weakly polar compounds [34].

CE-MS has been recognized recently as an attractive complementary technique for metabolomic studies and is particularly suitable for the separation of polar and ionic compounds based on a charge-to-mass ratio. The separation of CE is fast and highly efficient and does not need extensive sample pretreatment [35]. In addition, CE-MS only needs very low or even no organic solvents. A drawback of CE is the poor concentration sensitivity due to the limited sample volume. The currently available CE-MS techniques only allow sample loading of up to 1µL, and usually only utilize 10–100 nL [36]. In addition, migration times of metabolites can fluctuate with changing environmental temperatures, which can lead to reduced reproducibility [37].

DI-MS is a high throughput method with a short analysis time where the sample is directly introduced into the ESI source without chromatographic separation by using a syringe pump or nanospray chip [38]. However, its quantitative performance is inferior to LC-MS because of the strong matrix effect. A stable isotope labeling strategy has been applied to overcome the matrix effect [39].

NMR spectroscopy is an analytical technique based on the exploitation of the magnetic properties of atomic nuclei such as ^1^H, ^13^C, and ^31^P, allowing the identification of different atomic nuclei based on their resonant frequencies, which are dependent on their location in the molecule [14,24,40]. The NMR technique can uniquely and simultaneously quantify a wide range of organic compounds as well as provide unbiased information about metabolic profiles [29]. The applications of NMR spectroscopy are not only limited to liquid samples [41,42,43,44] but can also be used on solid [45,46] and tissue samples [47,48,49,50]. This platform is straightforward, largely automated, and non-destructive, so samples can be reused for further studies [15,29]. The major limitation of NMR for comprehensive metabolite profiling is its relatively low sensitivity, which makes it inappropriate for analyzing low-abundance metabolites [29].

Metabolomics techniques can be divided into untargeted and targeted. Untargeted metabolomics is a global, unbiased analysis of all small-molecule metabolites within a biological system, under a given set of conditions [51]. It measures hundreds of metabolites to identify metabolic changes, in a relative or non-quantitative way, and may serve to identify changed pathways for hypothesis building and further targeted studies [52]. Compared to targeted metabolomics, it is impossible to quantify all metabolites in untargeted metabolomics due to the large number of variables as well as the identity of metabolites is often unknown [53,54,55]. An important advantage of the untargeted approach is that it may also identify new metabolism areas [56,57]. The principal challenges of untargeted metabolomics lie in several aspects: (i) the protocols and time required to process the generated a large amount of raw data, (ii) the bias towards detection of molecules in high-abundance, (iii) the reliance on the intrinsic analytical coverage of the platform used, (iv) identifying and characterizing unknown small molecules [58]. In contrast, targeted metabolomics is the (semi-)quantitative measurement of a predefined set of metabolites [7]. It is commonly driven by a hypothesis or a specific biochemical question [59]. Targeted metabolomics can be effectively used for a pharmacokinetic study of drug metabolism as well as for measuring the influence of therapeutics or genetic modifications on specific enzymes [60].

There are many public databases available for metabolomics studies, such as the Human Metabolome Database (HMDB), METLIN, PubChem, and the Kyoto Encyclopedia of Genes and Genomes (KEGG) [61]. MetaboAnalyst (https://www.metaboanalyst.ca/) (accessed on 1 October 2022) is a powerful tool designed for processing and analyzing LC-MS-based global metabolomics data including spectral processing, functional interpretation, statistical analysis with complex metadata, and multi-omics integration [62].

### 2.2. Metabolomics Studies Related to AD

To investigate the current status of knowledge on metabolomics and insights into AD, for this review, an advanced literature search was performed using the following words: “Alzheimer’s disease [Title] AND (metabolomics OR lipidomics)”, until October 2022. It gave 498 results. We included experimental articles which compared the results of metabolomics analysis performed in biological samples taken from controls and from pathological conditions, both in AD animal models and in AD subjects. Case reports, reviews, editorials, conference summaries, and communications articles were excluded.

#### 2.2.1. Metabolomics in AD Human Studies

We found 44 articles that reported the outcomes of metabolomics analyses on CSF, plasma, saliva, and brain tissue samples from human subjects (Table 1) [63,64,65,66,67,68,69,70,71,72,73,74,75,76,77,78,79,80,81,82,83,84,85,86,87,88,89,90,91,92,93,94,95,96,97,98,99,100,101,102,103,104,105,106]. Among them, 5 articles used CSF sample alone, 25 articles used plasma samples alone, 8 articles used brain samples alone, 4 articles used plasma along with CSF samples, 1 article used postmortem CSF samples, and 1 article used saliva samples alone. These human studies have used metabolomics to establish disease-related plasma, brain, and CSF metabolite differences between cognitively normal (CN) individuals, mild cognitive impairment (MCI), and AD patients as predictors of AD progression.

In humans, CSF is the only fluid that can be sampled from the CNS, and therefore CSF is often used to obtain some information about what is happening in the brain. The composition of CSF reflects, to some extent, the composition of the brain’s extracellular space environment, which is in close connection with the metabolic processes occurring in this fluid and in the brain cells [107]. Several studies have shown that high CSF-total tau (tTau) and CSF-hyperphosphorylated tau (pTau) levels were found in early AD [108,109]. It is hypothesized that the metabolome in CSF may be altered in MCI or AD [64]. For that reason, the number of metabolomics-based studies investigating CSF composition is rapidly increasing. However, the collection of CSF through a lumbar puncture procedure is invasive and can be painful. It requires a patient’s cooperation which may be challenging especially for elderly people [15].

Blood samples are collected more easily compared to CSF, and would reduce the need for expensive, invasive, and time-consuming tests [110]. However, the difficulty in developing blood-based biomarkers for AD is underscored by the often-unknown ability of the molecule to pass through blood–brain barrier (BBB) and the difficulty in directly linking peripheral markers with brain processes [15]. Nevertheless, it is generally stated that the BBB is disrupted, which will increase permeability, with aging and in AD. Moreover, BBB disruption worsens as cognitive impairment increases, which means the relationships between metabolite concentrations in blood and the brain are strengthened [111]. However, there needs to be a somewhat better specification on what is meant by BBB permeability and conclusions on altered BBB transport [112].

Below we describe the results that have been reported in the metabolomics studies and summarize them in Table 1.

Maffioli et al. [103] explored the metabolome of healthy (n = 20) and AD-affected (n = 23) individuals by performing an untargeted metabolomics analysis on hippocampal samples. They detected 126 metabolites in total; 13 and 11 were up- and down-regulated, respectively, when comparing HC with AD samples. Enrichment analysis revealed that the most significantly upregulated pathways in AD samples were Arg/Pro metabolism and the pentose phosphate pathway. In contrast, the most significantly downregulated ones in AD samples were Ala/Asp/Glu metabolism, pyruvate metabolism, glycolysis/gluconeogenesis, pyrimidine metabolism, and aminoacyl-tRNA biosynthesis. In addition, gender-specific hallmarks of AD were explored. They found women with AD display a decrease in the D-serine/total serine ratio compared with men with AD.

To investigate the association between fatty acid metabolism and AD, Snowden et al. [82] conducted an untargeted metabolomics study. The samples were brain tissue from 43 individuals (14 AD, 14 healthy controls (HC), and 15 asymptomatic AD), ranging from 57 to 95 years old. They found that lower tissue levels of linoleic acid, linolenic acid, eicosapentaenoic acid, oleic acid, and arachidonic acid were related to worse cognitive performance, whereas higher brain DHA levels were associated with poorer cognitive performance.

Three potential biomarkers (ornithine, uracil, lysine) were identified by CE-TOF-MS on plasma samples from 40 HC, 26 MCI, and 40 AD patients [104]. For ornithine, there were significant differences between the HC and AD groups and between the MCI and AD groups. For uracil and lysine, there were significant differences between the HC and AD groups. They also measured mRNA expression levels of the metabolic enzymes in ornithine pathways, spermine synthase (SMS), nitric oxide synthase2 (NOS2), and ornithine transcarbamylase (OTC) mRNA levels were significantly different among the three groups.

Gonzalez-Dominguez et al. [68] utilized CE-TOF-MS to discover the early diagnostic biomarkers of Alzheimer’s disease. The serum samples were obtained from different stages of the disease (42 AD, 14 MCI, 37 HC). They found that with the progression of the disease, the levels of choline, creatinine, asymmetric dimethyl-arginine, homocysteine-cysteine disulfide, phenylalanyl-phenylalanine, and different medium-chain acylcarnitines were observed to increase significantly, while asparagine, methionine, histidine, carnitine, acetyl-spermidine, and C5-carnitine levels were reduced. Those metabolites are related to oxidative stress and defects in energy metabolism.

Shao et al. [96] used RP-UPLC-MS based untargeted metabolomics to measure the concentration of plasma metabolites among AD (n = 44) and cognitively normal control (n = 94) groups. Then, another cohort (43 HC, 31 neurological disease controls, 30 AD) was used to validate the result. They identified five metabolites that were able to distinguish AD patients from the HC group, which are allocholic acid, cholic acid, chenodeoxycholic acid, indolelactic acid, and tryptophan. This finding suggested that altered bile acid profiles in AD and MCI might indicate an early risk for AD development.

Wang et al. [71] applied UPLC-MS and GC-MS to analyze plasma samples from HC, MCI, and AD patients. They found a biomarker panel consisting of six metabolites (glutamine, glutamic acid, arachidonic acid, N,N-dimethylglycine, thymine, and cytidine) that can discriminate AD patients from control. Another panel of five metabolites (arachidonic acid, N,N-dimethylglycine, 2-aminoadipic acid, thymine, and 5,8-tetradecadienoic acid) was able to differentiate MCI patients from control subjects.

Peña-Bautista et al. [105] performed an untargeted lipidomics analysis on plasma samples from 20 healthy participants, 31 MCI-AD, and 11 preclinical AD. Statistically significant differences in the levels of Cer, lysophosphatidylethanolamine (LPE), lysophosphatidylcholine (LPC), and monoglyceride (MG) were observed between the preclinical AD and healthy groups. Statistically significant differences were also observed in the levels of diglycerol (DG), MG, and phosphatidylethanolamines (PE) between healthy groups and MCI-AD. In addition, LPE (18:1) showed significant differences between healthy participants and early AD (MCI and preclinical).

To determine whether the lipidome in AD has racial and ethnic disparities, Khan et al. [102] conducted a targeted lipidomics analysis of plasma samples from 54 HC and 59 AD from African American/Black (n = 56) and non-Hispanic White (n = 57) backgrounds. Five lipids (PS (18:0/18:0), PS (18:0/20:0), PC (16:0/22:6), PC (18:0/22:6), and PS (18:1/22:6)) were altered between the AD and HC sample groups. As for racial analysis, PS (20:0/20:1) was found reduced in AD in samples from non-Hispanic White but not altered in African American/Black samples.

Chouraki et al. [79] conducted a longitudinal assessment for 2067 participants with an average period of 15.6 ± 5.2 years to identify the novel biomarkers association with AD. Among 2067 participants, 93 developed dementia, including 68 with AD. Plasma samples were collected every 4 to 8 years. Four candidate plasma biomarkers were found for dementia through the metabolomics technique. Anthranilic acid, glutamic acid, taurine, and hypoxanthine levels were found to be associated with the risk of dementia.

Van der Lee [87] studied 299 metabolites in two discovery cohorts (n = 55,658) to find the associations with cognition. A total of 15 metabolites were discovered and replicated associated with cognition including subfractions of high-density lipoprotein (HDL), docosahexaenoic acid, ornithine, glutamine, and glycoprotein acetyls. Moreover, fish (oil) intake was found to be strongly associated with DHA blood concentrations (*p* = 9.9 × 10^−53^). Physical activity was found to be associated with increased (*p* < 0.05) levels of metabolites that were associated with higher cognitive function (medium and large HDL subfractions) and decreased levels of metabolites that were associated with lower cognitive function (glycoprotein acetyls, ornithine, and glutamine). Smokers were found to have decreased concentrations of all HDL subfractions associated with higher cognitive function and increased concentrations of metabolites associated with decreased cognitive function.

The Alzheimer’s Disease Neuroimaging Initiative (ADNI) began in 2004, and unites researchers who are investigating longitudinal data in AD. Here, clinical, neuroimaging, cognitive, biofluid biomarkers, and genetic data were collected to define the progression of AD [113]. Horgusluoglu et al. [101] systematically interrogated metabolomic, genetic, transcriptomic, proteomic, and clinical data from ADNI and found that short-chain acylcarnitines/amino acids and medium/long-chain acylcarnitines are most associated with AD clinical severity. Arnold et al. [114] used metabolomics data from 1571 participants of ADNI to investigate AD group-specific metabolic alterations. Fifteen metabolites were found to be associated with the female sex and APOE ε4 genotype: For CSF Aβ_1–42_, threonine showed a sex-specific effect with a greater effect size in males, while valine showed a larger effect in females. For CSF p-tau, acylcarnitines C5-DC (C6-OH), C8, C10, C2, and histidine showed stronger associations in females, whereas the related ether-containing PCs, PC ae C36:1, PC ae C36:2, asparagine, glycine, and one hydroxy-SM (SM (OH) C16:1) yielded stronger associations in males. MahmoudianDehkordi et al. [90] measured 15 primary and secondary bile acids in serum levels of 1464 subjects (37 CN older adults, 284 early mild cognitive impaired patients, 505 late mild cognitive impaired patients, and 305 AD). Primary bile acid cholic acid was found to be significantly lower serum levels in AD patients compared to CN subjects, whereas higher levels of secondary bile acids deoxycholic acid (DCA) and its conjugated forms (glycodeoxycholic acid (GDCA), glycolithocholic acid (GLCA), and taurolithocholic acid (TLCA)) were significantly associated with worse cognitive function.

Taken together, metabolomics allows the detection of metabolic alterations by monitoring multiple metabolites simultaneously. Multiple human studies have used metabolomics to distinguish age and sex-specific changes in plasma, brain, or CSF samples between CN subjects, MCI, and AD patients as predictors of AD progression which were then tested in animal models of AD to identify possible underlying causal mechanisms.

#### 2.2.2. Metabolomics in Animal Models of AD

More insights have been obtained on changes in metabolic pathways in AD, however, the lack of substantial time course data in humans is hindering understanding the sequence of disease stage-dependent changes in metabolic pathways. In other words, if we do not follow the change of metabolites and lipids over time, we cannot understand what changes along with disease progression and what may be important for developing adequate (stage-dependent) treatment of AD. This knowledge gap in biomarkers indicates that the onset and early stage of AD cannot be bridged by human studies since extensively obtaining human samples is by far more costly and time-consuming than obtaining samples from animals, it is also impossible to obtain human brain samples in longitudinal studies from the human aging population. Therefore, there is a need for alternative approaches to obtain the relation between the changes in biomarkers and AD stages, which can be achieved through animal model studies [115].

The most commonly used experimental animal models are transgenic mice that express human genes associated with familial AD (FAD) that result in the formation of amyloid plaques (by expression of human APP alone or in combination with human PSEN1), whereas human familial AD accounts for only 5% of cases [115,116,117,118,119]. Though not ideal, animal models provide an opportunity to study the early pathological disease mechanisms that can help to unravel processes associated with the development of AD. Additionally, animal models allow the investigation of (brain) tissues and fluids, and longitudinal studies can be performed to track disease progression, which cannot be accomplished in humans. Currently, along with the popular animal models of FAD including APP (Tg2576), APP/PS1, or 3xTg AD mice, the development of humanized mouse models expressing genetic risk factors, such as APOE ε4 allele, allows researchers to study mechanisms of late-onset sporadic AD [120,121,122].

In the present review, we specifically summarized AD mouse model research as only mouse research included age information in young mice. As summarized in Table 2, we found 42 articles [123,124,125,126,127,128,129,130,131,132,133,134,135,136,137,138,139,140,141,142,143,144,145,146,147,148,149,150,151,152,153,154,155,156,157,158,159,160,161,162,163,164] that carried out metabolomics analyses on samples of the brain, plasma, feces, spleen, and pancreas in mouse models of AD. Among them, nine articles studied plasma samples. A total of 21 articles only studied brain samples, and 9 articles conducted metabolomics experiments on plasma and brain samples. One study profiled spleen and brain samples, one study used serum and pancreas samples, one study used pancreas and serum samples, and one article used serum, brain, and feces samples on AD research. The results are described below.

Speers et al. [157] conducted metabolomics analysis of cortical tissue in 8-month-old male and female 5xFAD mice and their aged-match wild-type littermates. Sex differences were observed: 12 metabolites (betaine, lysine, pyridoxamine, urate, NAD, erythritol, spermine, N-acetylmannosamine, glycerol 2-phosphate, lauroyl-L-carnitine, sodium taurocholate, nicotinamide hypoxanthine dinucleotide) were significantly altered by the transgene in the female 5xFAD mice, whereas only five (homocysteine, betaine, N-Acetyl-mannosamine, S-Adenosyl-homocysteine, Adenosine 3′,5′-cyclic monophosphate) significantly altered metabolites in male 5xFAD.

Zhao et al. [159] applied targeted metabolomics on the hippocampi of 2- and 6-month-old triple transgenic AD male mice and age-sex-matched wild-type mice (WT). A total of 70 differential metabolites were identified, among them 24 metabolites were found changed in 2-month-old AD mice compared to WT, 60 metabolites were found changed in 6-month-old AD mice compared to WT. Fourteen metabolites were found in common, which are 7-methylguanosine, adenosine, adenosine 3′,5′-cyclic monophosphate (cAMP), cis-4-Hydroxy-d-proline, deoxycytidine, cytidine, deoxyadenosine monophosphate (dAMP), ethanolamine, glycerophosphocholine (GPC), L-2-aminoadipic acid, L-methionine, N-acetyl-D-glucosamine (GlcNAc), N-acetyl-L-tyrosine, and riboflavin (VB2). These results highlight the involvement of abnormal purine, pyrimidine, arginine, and proline metabolism, along with glycerophospholipid metabolism in the early pathology of AD.

Dejakaisaya et al. [156] identified alterations in cerebral metabolites and metabolic pathways in cortex, hippocampus, and serum samples from the Tg2576 AD mice model. Eleven metabolites showed significant differences in the cortex, including hydroxyphenyllactate (linked to oxidative stress) and phosphatidylserine (linked to lipid metabolism). For the network analysis, the authors used weighted correlation network analysis (WGCNA) to investigate the metabolite-group corrections. They identified five pathways, including alanine, aspartate, and glutamate metabolism, and mitochondria electron transport chain, that were significantly correlated with AD genotype.

Kim et al. [150] used untargeted metabolomics to investigate alterations in metabolite profiles of hippocampal tissues in 6-, 8- and 12-month-old wild-type and 5xfamiliar AD (5xFAD) mice. They found nicotinamide and adenosine monophosphate levels significantly decreased while lysophosphatidylcholine (LysoPC) (16:0), LysoPC (18:0), and lysophosphatidylethanolamine (LysoPE) (16:0) levels significantly increased in the hippocampi from 5xFAD mice at 8 months or 12 months of age when compared to age-matched wild type mice. In addition, the authors assumed that the primary neurons from 5xFAD reflect the hippocampal pathophysiological characteristics of 5xFAD. They treated the primary neurons with nicotinamide and found that treatment with nicotinamide rescued synaptic deficits in hippocampal primary neurons derived from 5xFAD mice. This finding indicated that decreased hippocampal nicotinamide levels could be linked with AD pathogenesis.

In 2020, Hunsberger et al. [149] collected prefrontal cortex, hippocampus, and spleen samples in 6-, 12- and 24-month-old APP/PS1 mice and age-matched wild-type mice. They conducted untargeted metabolomics analysis to investigate metabolomic alterations in naturally aged and APP/PS1 (AD) mice. Pathway analysis of changed metabolites revealed that across age, histidine metabolism was affected in all tissue samples, whereas amino acid metabolism and energy metabolism were altered in the prefrontal cortex, and AD significantly altered protein synthesis and oxidative stress in the hippocampus. Moreover, they found age-related metabolic changes occur earlier in the spleen compared to the CNS.

Zheng et al. [141] explored metabolic changes in six different brain regions between transgenic APP/PS1 mice and wild-type mice at 1, 5, and 10 months of age by using an NMR-based metabolomics approach to explore the metabolic mechanism that underlies the progression of amyloid pathology. They found the concentrations of glycerolphosphorylcholine, phosphocholine, and myo-inositol increased significantly in the hypothalamus of APP/PS1 mice when compared to WT mice, which indicated that the hypothalamus may be the main hypermetabolic region in the brain.

In conclusion, considering that biochemical pathways are largely conserved between humans and rodents [165], animal research is considered a valuable addition to human studies as human samples are costly and—especially in the case of brain samples—not available for longitudinal study. Different animal models of AD closely mimic the changes in metabolic networks associated with disease progression in humans.

## 3. AD Metabolic Pathways Analysis

MetaboAnalyst 5.0 (https://www.metaboanalyst.ca/) (accessed on 1 October 2022) is a website that provides software to analyze metabolomics data. It processes raw MS spectra, normalizes comprehensive data, and provides statistical analysis, functional analysis, meta-analysis, and metabolic pathway analysis. Metabolic pathway analysis identifies which metabolic pathways have compounds (from the user’s input list) that are over-represented and returns pathway impact. Pathway impact is calculated as the sum of the importance measures of matched metabolites normalized by the sum of the importance measures of all metabolites in each pathway [166]. Using this software, we analyzed the metabolic pathways to identify altered metabolites in the brain, plasma, and CSF in the literature studies described above.

The metabolic pathways were represented as circles according to their scores from enrichment (y-axis) and topology analyses (pathway impact, x-axis). Darker circle colors indicated more significant changes in metabolites in the corresponding pathway. The size of the circle corresponds to the pathway impact score and was correlated with the centrality of the involved metabolites.

### 3.1. Metabolic Pathway Analysis among Human Studies

After inputting all the altered metabolites from 43 human studies among different AD stages (MCI and AD) in brain, plasma, and CSF samples, 13 main metabolic pathways had a *p*-value less than 0.05 and impact value greater than 0.5 (Figure 1), including phenylalanine, tyrosine and tryptophan biosynthesis, taurine and hypotaurine metabolism, alanine, aspartate and glutamate metabolism, cysteine and methionine metabolism, arginine and proline metabolism, phenylalanine metabolism, tryptophan metabolism, arginine biosynthesis, beta-alanine metabolism, histidine metabolism, tyrosine metabolism, glycine, serine and threonine metabolism, and D-glutamine and D-glutamate metabolism. Details of pathway information are presented in Appendix A.

### 3.2. Common Regulated Metabolic Pathways among Human Plasma and CSF Sample

Furthermore, we investigated the altered metabolic pathways in common among different sample types in MCI and AD patients. After inputting all the altered metabolites from human MCI plasma samples, 12 metabolic pathways had a *p*-value less than 0.05 and impact values greater than 0. Using all the altered metabolites from the human AD plasma sample, 16 metabolic pathways had a *p*-value less than 0.05 and an impact value greater than 0. As shown in Figure 2A, a total of eight pathways were shared in the comparison between MCI VS. CN and AD VS. CN in plasma samples, which indicated that the metabolic mechanisms of AD and MCI share similar pathological alterations.

Similarly, all the altered metabolites from CSF samples were analyzed among MCI and AD groups. As shown in Figure 2B, a total of five pathways were shared in the comparison between MCI VS. CN and AD VS. CN. It is noted that all the MCI pathways overlapped with AD pathways in CSF samples.

We next used the combined (MCI + AD) altered metabolites found in plasma and CSF samples to understand important pathways in common between these two sample matrices. As shown in Figure 2C, a total of 13 pathways were shared between the plasma and CSF samples. Details of pathway information are presented in Appendix A.

### 3.3. Significantly Altered Metabolic Pathways among AD Mouse Models

For all the altered metabolites from the 35 mouse studies among different ages (2 months–24 months) in brain and plasma samples, when comparing to the control group, 13 main metabolic pathways were found with a *p*-value less than 0.05 and an impact value greater than 0.5 (Figure 3), including phenylalanine, tyrosine and tryptophan biosynthesis, linoleic acid metabolism, synthesis and degradation of ketone bodies, alanine, aspartate and glutamate metabolism, glycine, serine and threonine metabolism, arachidonic acid metabolism, phenylalanine metabolism, beta-alanine metabolism, arginine biosynthesis, glycerophospholipid metabolism, histidine metabolism, arginine and proline metabolism, and glyoxylate and dicarboxylate metabolism. Details of pathway information are presented in Appendix A.

### 3.4. Common Regulated Metabolic Pathways among Mouse Plasma and Brain Sample

Literature studies were combined to find disturbed pathways at different ages of AD mouse models, using all the altered metabolites at different ages in plasma and brain samples to perform pathway analysis. We included the pathways that meet a *p*-value less than 0.05 and an impact value greater than 0. The results showed that 31 pathways were significantly altered in mouse brain and plasma samples, the pathway impact values are shown as a heatmap (Figure 4A).

Furthermore, we investigated the altered metabolic pathways in common including all ages in plasma and brain samples. After including all the altered metabolites from mouse plasma samples, 18 metabolic pathways were found with a *p*-value less than 0.05 and an impact value greater than 0. Using all the altered metabolites from mouse brain samples, 18 metabolic pathways had a *p*-value less than 0.05 and an impact value greater than 0. As shown in Figure 4B, a total of 10 pathways were overlapping between the brain and plasma, indicating that the metabolic mechanisms seen in mouse plasma and brain share similar pathological alterations. Details of pathway information are presented in Appendix A.

## 4. Main Metabolic Pathways and Main Lipid Species with Respect to AD

The previous Section 3.1 and Section 3.3 highlighted that there was a total of 13 significant (*p*-value < 0.05 and impact value > 0.5) metabolic pathways altered in all matrices measured across all studies, including human and mouse research. Of these, eight altered metabolic pathways were found to be in common between AD mouse models and human AD subjects over all matrices: (i) alanine, aspartate, and glutamate metabolism, (ii) arginine and proline metabolism, (iii) arginine biosynthesis, (iv) β-alanine metabolism, (v) glycine, serine, and threonine metabolism, (vi) phenylalanine metabolism, vii) histidine metabolism, and (viii) phenylalanine, tyrosine, and tryptophan biosynthesis. Section 3.2 highlighted metabolic pathways in human plasma and CSF samples. Section 3.4 highlighted metabolic pathways in mouse brain and plasma samples. The matrix commonly investigated in both human and mouse research is plasma. There were 17 significant (*p*-value < 0.05 and impact value > 0.5) metabolic pathways in human plasma samples, whereas there were 18 significant (*p*-value < 0.05 and impact value > 0.5) metabolic pathways in mouse plasma samples. Of all of these, there were 12 metabolic pathways that were altered in both AD mouse models and human AD subjects in plasma: (i) alanine, aspartate, and glutamate metabolism, (ii) arginine and proline metabolism, (iii) arginine biosynthesis, (iv) butanoate metabolism, (v) citrate cycle (TCA cycle), (vi) glutathione metabolism, (vii) glycerophospholipid metabolism, (viii) glycine, serine, and threonine metabolism, (ix) glyoxylate and dicarboxylate metabolism, (x) linoleic acid metabolism, (xi) phenylalanine metabolism, and (xii) phenylalanine, tyrosine, and tryptophan biosynthesis. The following section is a detailed description of these important metabolic pathways that emerged as significantly altered after consolidating the analysis results.

### 4.1. Arginine Metabolism

L-arginine is a semi-essential amino acid that can be metabolized to form a number of bioactive molecules [167] (Figure 5). It is synthesized from proline or glutamate, with the ultimate synthetic step catalyzed by argininosuccinate lyase [168]. L-arginine can be metabolized by arginases, nitric oxide synthases (NOS), and possibly also by arginine decarboxylase (ADC), resulting ultimately in the production of agmatine, ornithine, nitric oxide (NO), or urea [168]. The expression of several of these enzymes can be regulated at transcriptional and translational levels by changes in the concentration of L-arginine itself [169].

L-ornithine is the arginase-mediated metabolite of L-arginine, with urea as the by-product. L-ornithine can be further metabolized to form putrescine, spermidine, and spermine polyamine, which are essential for normal cell growth and functioning, or via a separate pathway to form glutamine and cell-signaling molecule, GABA [167]. Previous research has reported decreased glutamate and GABA levels in AD brains and increased glutamine synthase (GS) levels in the lumbar cerebrospinal fluid of AD patients [170,171]. In peripheral organs and also CNS, arginine can also be metabolized by ADC to produce agmatine, a neurotransmitter that plays an important role in the learning and memory process [172].

NO is a gaseous signaling molecule produced by NOS. NO, derived from neuronal NOS (nNOS), plays an important role in synaptic plasticity and learning, and memory [173,174,175]. Moreover, L-arginine and NO affect the cardiovascular system as endogenous antiatherogenic molecules that protect the endothelium, modulate vasodilatation, and interact with the vascular wall and circulating blood cells [176,177,178,179,180].

### 4.2. Alanine, Aspartate, and Glutamate Metabolism

Glutamate is the principal excitatory neurotransmitter of the brain [181]. Most neurons and glia are likely to be influenced by glutamate since they have receptors for glutamate. Glutamate is considered the main neurotransmitter of neocortical and hippocampal pyramidal neurons and is involved in higher mental functions such as cognition and memory [182]. Disturbance of excitatory glutamatergic neurotransmission is believed to be associated with many neurological disorders, including Alzheimer’s disease (AD) [182], ischemic brain damage [183], and motor neuron disease [184].

Glutamate receptors can be divided into two classes: ionotropic (N-methyl-D-aspartate, NMDA) and α-amino-3-hydroxy-5-methyl-4-isoxazolepropionic acid (AMPA)/kainite subtypes and metabotropic [185]. The role of glutamate and glutamate receptors in learning and memory is widely recognized. For instance, NMDA antagonists impair learning and memory while NMDA agonists and facilitators improve memory [182]; likewise, AMPAKines (positive modulators of receptor function) facilitate learning and memory [186]. Circumstantial evidence of the involvement of glutamatergic pathways derives from the well-known role of structures such as the hippocampus in learning and memory [187]. More specifically, lesions of certain glutamatergic pathways impair learning and memory [188]. Moreover, glutamate and glutamate receptors are involved in mechanisms of synaptic plasticity, which are considered to underlie learning and memory [189,190,191].

### 4.3. Purine Metabolism

Purines and pyrimidines are components of many key molecules in living organisms. The primary purines adenine and guanosine and the pyrimidines cytosine, thymidine, and uracyl are the core of DNA, RNA, nucleosides, and nucleotides involved in energy transfer (ATP, GTP) [192,193]. Several studies indirectly suggested that purine metabolism has altered in AD. Energy metabolism, which depends on mitochondrial function and ATP production, is markedly altered in AD [194,195]. In addition, oxidative damage to DNA and RNA, as revealed by the increase in 8-hydroxyguanosine, is found in the brain samples of AD [196,197,198,199]. Direct alterations of purine metabolism in AD have been detected by metabolomics in postmortem ventricular CSF [200] and in the spinal cord CSF of living individuals [201,202,203]. Only a limited number of metabolomics studies have been carried out in AD brains [203].

### 4.4. Taurine and Hypotaurine Metabolism

Taurine is the second most abundant endogenous amino acid in the central nervous system (CNS) and has multiple roles in our body: thermoregulation [204], stabilization in regulating protein folding [205], anti-inflammatory effects [206], antioxidation [207], osmoregulation [208], and calcium homeostasis [209]. Recently, taurine has shown therapeutic effects as a cognitive enhancer in animal models of non-AD neurological disorders [210,211,212,213]. Taurine protected mice from the memory disruption induced by alcohol, pentobarbital, sodium nitrite, and cycloheximide but had no obvious effect on other behaviors including motor coordination, exploratory activity, and locomotor activity [210]. Intravenously injected taurine significantly improves post-injury functional impairments of traumatic brain injury in rats [211]. The intracerebroventricular (ICV) administration of taurine protects mice from learning impairment induced by hypoxia. Neither beta-alanine nor saccharose was able to mimic the effects of taurine [212]. In streptozotocin-induced sporadic dementia rat models, cognitive impairment and deterioration of neurobehavioral activities are ameliorated by taurine [213].

Taurine also has multiple disease-modifying roles to cease or prevent AD neuropathology. During the development of AD, amyloid-β (Aβ) progressively misfolded into toxic aggregates, which are strongly associated with neuronal loss, synaptic damage, and brain atrophy. An electron microscopy study indicates that taurine slightly decreases β-amyloid peptide aggregation in the brain at a millimolar concentration [214]. Taurine also has anti-inflammatory and antioxidant properties; it can provide protection for neuronal cells and mitochondria from the neurotoxicity of Aβ. By activating GABA and glycine receptors, taurine inhibits excitotoxicity caused by Aβ-induced glutamatergic transmission activation [215].

### 4.5. Cholinergic System

As acetylcholine (ACh) plays a vital role in cognitive processes, the cholinergic system is considered an important factor in AD [216]. The brain regions most affected by a loss of elements of the acetylcholine system include the hippocampus, cortex, and entorhinal [217]. Cholinesterase inhibitors are one of the few drug therapies available in the clinic for the treatment of AD, and it was inspired by the fact that cholinesterase inhibitors increase the availability of acetylcholine at brain synapses [218]. The validation of the cholinergic system was seen as an important therapeutic target in the disease.

### 4.6. Fatty Acids

Fatty acids are the basic building blocks of more complex lipids and can be classified by the number of double bonds as saturated fatty acids (SFAs) and unsaturated fatty acids. SFAs do not include any double bonds, whereas unsaturated fatty acids contain at least one (monounsaturated fatty acids, MUFAs) or two or more (polyunsaturated fatty acids, PUFAs) double bonds [219,220]. Altered unsaturated fatty acids have been associated with AD in multiple studies. The brain is especially enriched with two PUFAs: docosahexaenoic acid (DHA) and arachidonic acid (AA). DHA, as one of omega-3 PUFAs, is the predominant structural fatty acid in the mammalian brain and plays an essential role in brain functioning, especially in cognitive function; DHA levels were lower in AD brains [221,222] or plasma [69], and increased intake of DHA from fish or marine oils may lower AD risk [223,224,225]. AA of the ω-6 fatty acid family appears to play critical mediator roles in amyloid (Aβ)-induced pathogenesis, leading to learning, memory, and behavioral impairments in AD [226]. The levels of free AA have been found to increase in AD patient brain samples [82], whereas the levels of AA in phospholipids are reduced in the hippocampus of AD subjects [227].

### 4.7. Glycerolipids

Glycerolipids can be categorized into triacylglycerols (TAG, also known as triglycerides, TG), monoacylglycerol (MAG), and diacylglycerol (DAG) based on the number of acyl groups in the structure. TAG, the most predominant glycerolipids, are esters composed of a glycerol backbone and three fatty acids. TAG levels are found not to be changed in the serum of AD patients when compared to control subjects. [228]. However, MAG and DAG are elevated in both the prefrontal cortex and plasma of AD and MCI subjects in comparison to controls [229,230]. Moreover, MAG and DAG are elevated in the grey matter of MCI and AD patients, suggesting that these biochemical changes may play a role in the development of MCI and in the transition from MCI to AD [231].

### 4.8. Glycerophospholipids

Glycerophospholipids (GPs), also referred to as phospholipids (PLs), are typically amphipathic and make up the characteristic lipid bilayer structure of biological membranes. Moreover, GPs are the major type of lipids that make up cell membranes and account for 50–60% of the total membrane mass along with cholesterol and glycolipids [232]. GPs include phosphatidylethanolamine (PE), phosphatidic acid (PA), phosphatidylserine (PS), phosphatidylglycerol (PG), phosphatiylcholine (PC), phosphatidylinositol (PI), sphingomyelin (SM), and cardiolipin (CL) [233]. Studies on GP composition indicate that levels of PC, PE, and PI are significantly decreased in neural membranes from different regions of AD patients compared to age-matched control brains [234,235,236,237,238,239].

Phosphatidylethanolamine (PE) is converted to lysophosphatidylethanolamine (lyso-PE) by phospholipase A2 (PLA2), an important inflammatory mediator that is dysregulated in AD. PLA2 level has been found to be elevated in the human cerebral cortex [240] or decreased in the human parietal and frontal cortex [241]. Moreover, PLA2 influences the processing and secretion of amyloid precursor protein, which gives rise to the β-amyloid peptide, the major component of the amyloid plaque in AD [241]. Moreover, PLA2 has been found to play an important role in memory retrieval [242].

Phosphatidylserine (PS) is the major acidic phospholipid class that accounts for 13–15% of the phospholipids in the human cerebral cortex [243]. PS is known as a “brain nutrient”, as it can not only nourish the brain, but also enhance brain functions such as improving cognition, memory, and reaction force [244]. In six double-blind trials, PS has been found effective for AD. At daily doses of 200–300 mg for up to six months, PS consistently improved clinical global impression and activities of daily living [245]. In milder cases, PS improved orientation, concentration, learning, and memory for names, locations, and recent events. In the largest trial, involving 425 elderly patients (aged between 65 and 93 years) with moderate to severe cognitive decline, PS significantly improved memory, learning motivation, and socialization, suggesting that it has a vital impact on the quality of life of such elderly patients.

Phosphatidylcholine (PC) is an essential component of cell membranes and makes up approximately 95% of the total choline compound pool in most tissues [246,247]. Its function is defined primarily by chain length since chain length differences can affect cell membrane fluidity [248]. Three PCs (PC 16:0/20:5, PC 16:0/22:6, and PC 18:0/22:6) have been found significantly diminished in AD patients [249].

Lysophosphatidic acids (LPAs) are phospholipids derivatives that can act as signaling molecules [250]. Ahmad et al. [95] investigated the association between LPAs and CSF biomarkers of AD, Aβ-42, p-tau, and total tau levels overall and with MCI to AD progression. Five LPAs (LPA C16:0, LPA C16:1, LPA C22:4, LPA C22:6, and isomer-LPA C 22:5) correlated significantly and positively with CSF biomarkers of AD, Aβ-42, p-tau, and total tau. Additionally, LPA C16:0 and LPA C16:1 showed associations with MCI to AD dementia progression.

### 4.9. Sphingolipids

Sphingolipids, a class of membrane biomolecules, include sphingosine 1-phosphates (S1P), Cers, SMs, and glycosphingolipids, which are vital for maintaining cell integrity and signal transduction processes [251]. Cers, the basic structural units of the sphingolipid class, have been seen as key contributors to the pathology of AD as they are able to affect both Aβ generation and tau phosphorylation [252]. Filippov et al. found elevated levels of ceramides Cer16, Cer18, Cer20, and Cer24 in the brains of AD patients. Two saturated ceramides, Cer (d18:1/18:0) and Cer (d18:1/20:0) were significantly increased in the senile plaques [253]. High ceramide levels were also found in AD serum [254] and CSF samples [255]. The greatest genetic risk factor for late-onset AD is the ε4 allele of apolipoprotein E (ApoE). ApoE regulates the secretion of the potent neuroprotective signaling lipid S1P [256]. S1P is derived by phosphorylation of sphingosine, catalyzed by sphingosine kinases 1 and 2 (SphK1 and 2). SphK1 positively regulates glutamate secretion and synaptic strength in hippocampal neurons. Reduced levels of S1P have been found in AD brains compared to controls [256,257]. All these studies mentioned above suggested that sphingolipid metabolism plays a critical role in AD pathology.

### 4.10. Cholesterol and Cholesteryl Esters

Despite the brain occupying only 2% of total body weight, it contains 25% of the body’s cholesterol. Due to the BBB, cholesterol metabolism in the CNS is largely separated from that in the periphery and cholesterol is de novo synthesized in the CNS [258]. Studies have found that brain cholesterol was significantly increased in AD patients than in controls [259,260]. Moreover, cholesterol showed abnormal accumulation in the senile plaques of the human brain, a hallmark neuropathological feature of AD [261].

## 5. Conclusions and Future Directions

Considering the dramatic aging of populations worldwide, it is of great importance to explore AD pathogenesis. Metabolic changes associated with AD progression occur prior to the development of clinical symptoms; metabolomics by itself or in conjunction with the additional currently available biomarkers for AD diagnosis could serve as an additional tool to increase the accuracy of diagnosis, to predict the disease progression, and to monitor the efficacy of therapeutic intervention.

Metabolomic studies have demonstrated the dramatic impact of AD pathogenesis and progression on metabolites and related metabolic pathways, including energy-related metabolism, fatty acid metabolism, abnormal lipid metabolism, altered amino acids metabolism (e.g., arginine, glutamate), and some others. In the present review, we summarized the metabolomics studies that were performed in biological samples of AD subjects and AD mouse models. The results of rats were too sparse and not suitable for further analysis. The mouse research shows that 12- and 24-months, middle and old age in AD mouse models, can be equivalated to the MCI and AD late stage in humans, respectively. As obtaining human body samples is costly, limited in possible samples sites by ethics (i.e., brain), and time-consuming for the long life span of humans, the above indicates that animal research may be considered a valuable addition, as it can be designed in a longitudinal fashion and with samples from multiple sites of the body to obtain time-course information and interrelationships, to gain insights that support research on AD in humans.

The disturbed pathways by AD were analyzed based on metabolite data collected from the literature. Eight disturbed metabolic pathways were found in common between AD mouse research and AD human research. These pathways are alanine, aspartate, and glutamate metabolism, arginine and proline metabolism, arginine biosynthesis, β-Alanine metabolism, glycine, serine and threonine metabolism, phenylalanine metabolism, histidine metabolism, phenylalanine, tyrosine, and tryptophan biosynthesis.

Our analysis of the literature studies has several limitations. The coverage of metabolites varied among different studies due to the detection sensitivity differences, as analytical platforms (e.g., NMR, GC-MS, LC-MS) and analytical methods are diverse in different laboratories. In addition, it is hard to reproduce various metabolomics results because of different sample sources (brain tissue or plasma) from either deceased or living patients and diverse distribution about sex, age, and suffering from other diseases. Moreover, the methods used for obtaining samples, such as CSF and brain, varied among different studies. For example, delays between removing and freezing animal or human brain tissue can affect metabolomics analysis.

Altogether, in this review, we summarized and analyzed existing metabolomics data and the relation between plasma, CSF, and brain for animals and plasma and CSF for human data. We identified missing longitudinal information, which would be difficult to be obtained from humans (high costs, long direction) while also in the human brain cannot be sampled. Longitudinal and multi-body site information, however, is important to understand the processes in AD. This is where animal research may support AD research in humans to provide new insights on disease biomarkers patterns and biological pathways, that will support AD stage diagnosis in humans but also the discovery of AD future therapeutic targets.

## Figures and Tables

**Figure 1 ijms-24-04960-f001:**
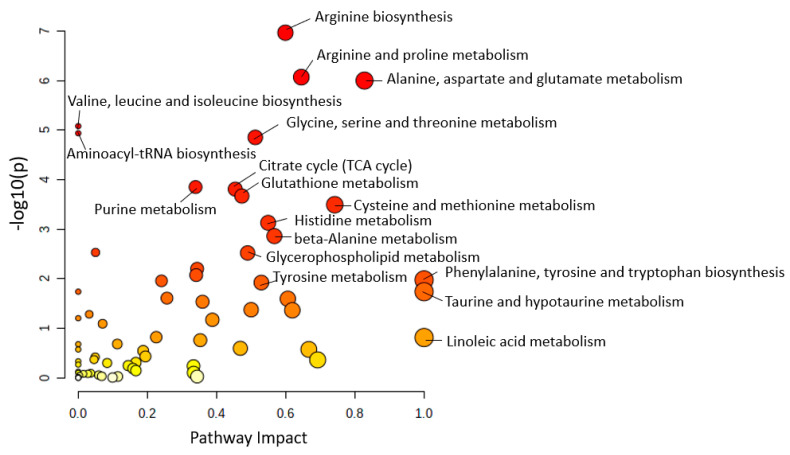
Metabolic pathway analysis in biological samples (CSF, brain, and plasma) of 43 human AD studies. The size of the circle corresponds to the pathway impact score and was correlated with the centrality of the involved metabolites. Darker circle colors indicated more significant changes in metabolites in the corresponding pathway.

**Figure 2 ijms-24-04960-f002:**
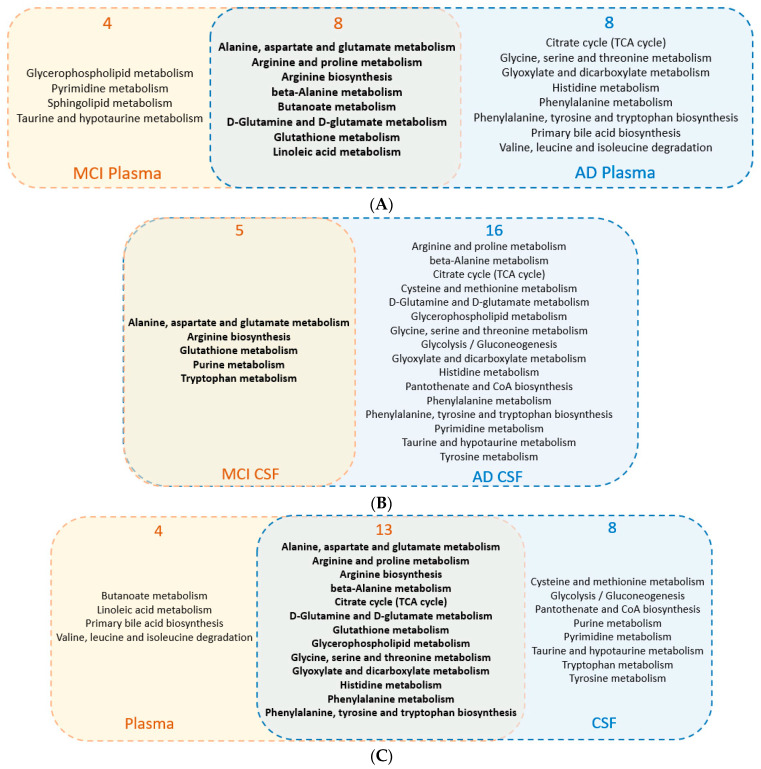
Disturbed metabolic pathways analysis in humans, comparing different disease stages with healthy controls: (**A**) Intersection analysis of metabolic pathways among MCI and AD groups in plasma samples. Yellow circles are all the disturbed pathways in MCI plasma samples, blue circles are all the disturbed pathways in AD plasma samples, and the middle intersection is pathways in common among MCI and AD in plasma samples. (**B**) Intersection analysis of metabolic pathways among MCI and AD groups in CSF samples. Yellow circles are all the disturbed pathways in MCI CSF samples, blue circles are all the disturbed pathways in AD CSF samples. All the MCI pathways overlapped with AD pathways in CSF samples. (**C**) Intersection analysis of metabolic pathways among plasma and CSF samples. Yellow circles are all the disturbed pathways in plasma samples, blue circles are all the disturbed pathways in CSF samples, and the middle intersection is pathways in common among plasma and CSF samples.

**Figure 3 ijms-24-04960-f003:**
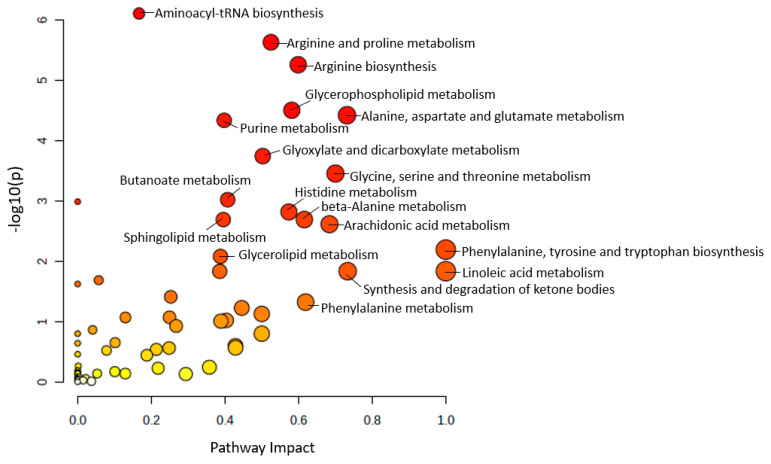
Metabolic pathway analysis in biological samples (brain and plasma) of 42 mouse studies. The size of the circle corresponds to the pathway impact score and was correlated with the centrality of the involved metabolites. Darker circle colors indicated more significant changes in metabolites in the corresponding pathway.

**Figure 4 ijms-24-04960-f004:**
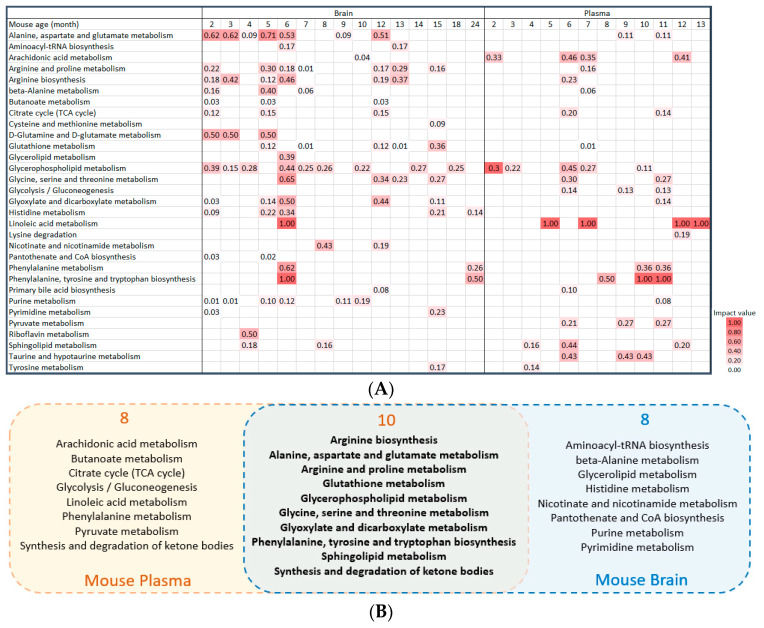
Altered metabolic pathways in mouse plasma and brain samples, comparing AD and control animals. (**A**) Heatmap of the changes of significantly altered metabolic pathways at different ages of AD mouse in brain and plasma samples. The impact value is calculated as the sum of the importance measures of matched metabolites normalized by the sum of the importance measures of all metabolites in each pathway. (**B**) Altered pathways in AD mouse plasma and brain samples combining all age groups. Yellow circles are all the disturbed pathways in mouse plasma samples, blue circles are all the disturbed pathways in mouse brain samples, and the middle intersection is pathways in common among plasma and brain samples in mouse research.

**Figure 5 ijms-24-04960-f005:**
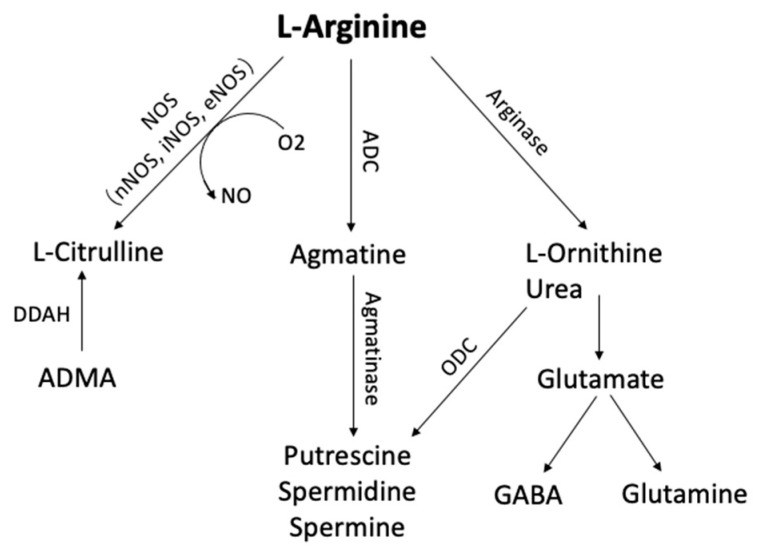
Arginine metabolic pathways. L-arginine can be metabolized by phosphatidic acid (PA), nitric oxide synthase (NOS), arginase, and arginine decarboxylase (ADC) to form several bioactive molecules. (ADC, Arginine decarboxylase; ADMA, NG-dimethyl-L-arginine; DDAH, dimethylarginine dimethylaminohydrolase; GABA, γ-aminobutyric acid; ODC, ornithine decarboxylase;).

**Table 1 ijms-24-04960-t001:** Summary of the included articles related to metabolomics studies performed on human subjects.

No.	Reference	Study Population	Sample Type	Analytical Platform	Altered Metabolites
1	Czech, et al., 2012 [63]	51 HC, 79 AD	CSF	LC-MS, GC-MS	Citrulline, Cortisol, Creatinine, Cysteine, Dopamine, Erythrol, Galactitol, Histamine, Methionine, Noradrenaline, Normetanephrine, Phenylalanine, Pseudouridine, Pyruvate (incl. Phosphoenolpyruvate), Quinic acid (incl. Chlorogenic acid), Ribonic acid, Scyllo-inositol, Serine, Sorbitol (incl. Mannitol, Galactitol), Tyrosine, Uridine, Ornithine (incl. Arginine, Citrulline)
2	Ibáñez, et al., 2012 [64]	19 HC, 22 MCI, 9 MCI-AD, 23 AD	CSF	CE-TOF-MS	Choline, Valine, Arginine, Tripeptide, Carnitine, Dimethy-L-arginine, Creatine
3	Sato, et al., 2012 [65]	10 HC, 10 AD	Plasma, CSF	LC/APCI-MS	Cholesterol, Desmosterol
4	Ibáñez, et al., 2013 [66]	21 HC, 21 MCI-S, 12 MCI-AD, 21 AD	CSF	UHPLC-TOF-MS	Uracil, Xanthine, Uridine, Tyrosyl-serine, Methylsalsolinol, Nonanoylglycine, Dopamine-quinone, Caproic acid, Vanylglycol, Histidine, Pipecolic acid, Hydroxyphosphinyl-pyruvate, Creatinine, Taurine, Sphingosine-1-phosphate, Tryptophan, Methylthioadenosine
5	Luliano, et al., 2013 [67]	30 HC, 14 MCI, 30 AD	Plasma	GC-MS	Arachidic acid, Cerotic acid, cis-Vaccenic acid, Erucic acid, Linoleic acid, Mead acid
6	González-Domínguez, et al., 2014 [68]	37 HC, 14 MCI, 42 AD	Plasma	CE-MS	Choline, Creatinine, Asparagine, Proline betaine, Methionine, Histidine, Carnitine, N-acetyl-spermidine, Asymmetric dimethyl-Arginine, Tripeptide
7	González-Domínguez, et al., 2014 [69]	18 HC, 22 AD	Plasma	DIMS/MS	Caprylic acid, Capric acid, Lauric acid, Myristic acid, Palmitoleic acid, Palmitic acid, Linoleic acid, Docosahexaenoic acid, Leukotriene B4, Prostaglandin, Choline, Valine, Creatine, Glutamine, Glutamate, Dopamine, Histidine, Carnitine, Arginine, N-acetyl glutamine, Glucose, Glycerophosphocholine, Lyso-phospholipids, Phospholipids
8	González-Domínguez, et al., 2014 [70]	18 HC, 22 AD	Plasma	ESI-Q-TOFMS	Arginine, Guanidine, Histidine, Imidazole, Kynurenine, Oleamide, P18:0/C22:6-PlsEt, P18:1/C20:4-PlsEt, Prostaglandins, Putrescine, Taurine
9	Wang, et al., 2014 [71]	57 HC, 58 MCI, 57 AD	Plasma	UPLC-QTOF-MS	Panel for MCI: Thymine, Arachidonic acid, 2-Aminoadipic acid, N,N-dimethylglycine, 5,8-Tetradecadienoic acid Panel for AD: Arachidonic acid, N,N-dimethylglycine, Thymine, Glutamine, Glutamic acid, Cytidine
10	González-Domínguez, et al., 2015 [72]	30 HC, 30 AD	Plasma	FIA-APPI-QTOFMS	Palmitoleamide, Palmitamide, Linolenamide, Linoleamide, Oleamide, Stearamide, Palmitoleic acid, Palmitic acid, Oleic acid, Urea, Alanine, Taurine, Picolinic acid, Creatine, Malic acid, Dopamine, Serotonin, Ceramides (Cers), Diacylglycerols (DAG)
11	González-Domínguez, et al., 2015 [73]	21 HC, 23 AD	Plasma	GC-MS	Adenosine, Asparagine, Aspartic acid, Cholesterol, Cystine, Glucose, Glutamine, Histidine, Isocitric acid, Lactic acid, Oleic acid, Ornithine, Palmitic acid, Phenylalanine, Pipecolic acid, Pyroglutamic acid, Stearic acid, Tryptophan, Tyrosine, Urea, Uric acid, Valine, α-Ketoglutarate
12	Graham, et al., 2015 [74]	37 HC, 16 MCI, 19 MCI-AD	Plasma	LC-QTOF-MS	4-Aminobutanal, Creatine, γ-aminobutyric acid (GABA), L-ornithine, N1-diacetlyspermine, N-acetylputrescine, Spermine, L-arginine, Methylthioadenosine, N1-acetyl-spermidine, Putrescine, Spermidine
13	González-Domínguez, et al., 2016 [75]	45 HC, 17 MCI, 75 AD	Plasma	UPLC-QTOF-MS	Lyso-phospholipids, Phospholipids, S1P, Cers, Sphingomyelins, Glycosphingolipids, Monoglycerides, Acyl-carnitines, Histidine, Phenyl-acetyl-glutamine, Oleamide
14	Paglia, et al., 2016 [76]	19 HC, 21 AD	Frontal cortex	UPLC-MS	Acetylaspartic acid, Acetylglutamic acid, ADP, ADP-ribose, Alanine, AMP, Arginine, Asparagine, Aspartic acid, Choline, Cystine, Glutamic acid, Glutamine, GMP, Guanosine, Hydroxyproline, Hypoxanthine, IMP, Inosine, Methionine, Pantothenic acid, Pentose 5-phosphate, Proline, Pyruvate, SAH, SAMe, Serine, Succinic acid, Threonine, Tryptophan, Uric acid, Valine, Xanthine, Xanthosine
15	Vaňková, et al., 2016 [77]	22 HC, 16 AD	Plasma	GC-MS	17-Hydroxypregnenolone, 17-Hydroxyprogesterone, 20α-Dihydroprogesterone, Allopregnanolone, Allopregnanolone sulfate, Androstenedione, Conjugated 5α-androstane-3b,17b-diol, Isopregnanolone, Pregnanolone, Pregnenolone sulfate, steroid
16	Xu, et al., 2016 [78]	9 HC, 9 AD	Brain	GC-MS	55 altered metabolites belonging to glucose utilization/clearance and brain energetics metabolism, and urea and amino-acid metabolism
17	Chouraki, et al., 2017 [79]	1974HC, 93AD	Plasma	LC-MS	Anthranilic acid, Glutamic acid, Taurine, Hypoxanthine
18	de Leeuw, et al., 2017 [80]	121 HC, 127 AD	Plasma	Multiple Mass spectrometry platforms	Tyrosine, Glycylglycine, Glutamine, Lysophosphatic acid C18:2
19	Oberacher, et al., 2017 [81]	18 HC, 15 MCI, 21 AD	Plasma	FIA-MS/MS	PCaa C32:0, PCaa C34:1, PCaa C36:5, PCaa C36:6, PCaa C38:0, PCaa C38:3, PCae C32:1, PCae C32:2, PCae C34:1, PCae C40:4, lysoPC aC16:0, lysoPC aC18:1, lysoPC aC18:2, SM (OH) C14:1
20	Snowden, et al., 2017 [82]	14 HC, 14 AD	Brain	LC-MS and GC-MS	Linoleic acid, Linolenic acid, Eicosapentaenoic acid, Oleic acid, Arachidonic acid
21	Hao, et al., 2018 [83]	14 HC, 16 AD	CSF	LC-MS/MS	142 metabolites belong to carboxylic acids, amino acids, fatty acyls, fatty acids and conjugates, pyrimidines, nucleosides, and analogs. E.g.: 3-Oxododecanoic acid, Dodecanedioic acid, Methylerythritol phosphate, Glutamine, Dihydrothioctic acid, Deoxyinosine, Succinyl-glutamate
22	Kim, et al., 2018 [84]	13 HC, 23 MCI, 14 AD	Plasma	UHPLC-ESI-MS/MS	PE (34:2), PE (36:2), PE (38:4), PE (38:5), PA (18:1/22:6), PI (18:0/20:4), PI (18:1/16:1), PI (18:2/16:1), Cer (d18:1/22:0), Cer (d18:1/24:1), HexCer (d18:1/24:0), DG (18:0/22:6), DG (18:1/18:1), TG (50:1)
23	Muguruma, et al., 2018 [85]	10 HC, 10 AD	Postmortem CSF (pCSF)	UHPLC-MS/MS	Polyamine and tryptophan-kynurenine (Trp-Kyn) metabolisms, such as methionine sulfoxide, 3-methoxy-anthranilate, cadaverine, guanine, and histamine
24	Nasaruddin, et al., 2018 [86]	27 HC, 16 AD	Brain	GC-MS	Arachidic acid, Arachidonic acid, Behenic acid, Cis-10-heptanoic acid, Cis-11,14,17-eicosatrienoic acid, Cis-13,16-docosadienoic acid, Erucic acid, Lignoceric acid, Linolenic acid, Nervonic acid, Oleic acid, Palmitic acid, Stearic acid
25	van der Lee, et al., 2018 [87]	23,882 HC, 1356 AD	Plasma	NMR spectroscopy	High-density lipoprotein subfractions, Docosahexaenoic acid, Ornithine, Glutamine, Glycoprotein acetyls
26	Kim, et al., 2019 [88]	9 HC, 9 AD	Postmortem, brain	UPLC-MS	Hypotaurine, Myo-inositol, Oxo-proline, Glutamate, N-acetyl-aspartate, Cortisol, N-acetylaspartate (NAA), N-acetylaspartylglutamate (NAAG), Acetylcholine, Alanine
27	Lin, et al., 2019 [89]	15 HC, 10 MCI, 15 AD	Plasma	LC-MS/MS	Propionylcarnitine, Valerylcarnitine, Glutarylcarnitine/Hydroxyhexanoylcarnitine, Arginine, Phenylalanine, Creatinine, Symmetric dimethylarginine (SDMA)
28	MahmoudianDehkordi, et al., 2019 [90]	37 HC, 284 early MCI, 505 late MCI, 305 AD	Plasma	UPLC-MS/MS	Cholic acid, Deoxycholic acid (DCA), Glycodeoxycholic acid (GDCA), Glycolithocholic acid (GLCA), Taurolithocholic acid (TLCA)
29	Marksteiner, et al., 2019 [91]	25 HC, 26 MCI, 27 AD	Saliva	FIA-MS/MS	PCae C34:1-2, PCae C36:1-2-3, PCaeC38:1-3, PCae C40:2-3, PCae C36: (1-2-3)
30	Peña-Bautista, et al., 2019 [92]	29 HC, 29 AD	Plasma	UPLC-Q-TOF-MS	Choline, L-carnitine, 4-Deoxyphysalolactone, Rescinnamine, Chlorohydrin, Brassinin, Nicotinamide ribotide, Cyasterone
31	Snowden, et al., 2019, [93]	14 HC, 14 AD	Brain	LC-MS	Aminobutanal, Arginine, Aspartate, Dihydroxy-phenylalanine, Dopamine, Gamma-aminobutanoate, Glutamate, Glycine, Guanidinobutanoate, Guanosine, Ornithine, Serotonin, Tryptophan, Tyrosine
32	van der Velpen, et al., 2019 [94]	34 HC, 40 AD	Plasma, CSF	LC-MS/MS	Acetylcarnitine, Acylcarnitines C14, Acylcarnitines C16, Acylcarnitines C18, cis-Aconitate, Citrate, Creatinine, Decanoylcarnitine, Hexanoylcarnitine, Kynurenic acid, Lauroylcarnitine, L-carnitine, Octanoylcarnitine, Quinolinic acid, Tryptophan, α-Ketoglutarate
33	Ahmad, et al., 2020 [95]	Two independent cohorts: 142 MCI and 40 MCI	CSF and plasma	UHPLC-MS/MS	LPA (C16:0), LPA (C16:1), LPA (C22:4), LPA (C22:6), and isomer-LPA (C22:5)
34	Shao, et al., 2020 [96]	94 HC, 44 AD	Plasma	UPLC-MS	Polyunsaturated fatty acids (PUFAs), Docosahexaenoic acid (DHA, C22:6), medium- and long-chain Acyl-Carnitines, Cholic acid (CA), Chenodeoxycholic acid (CDCA), Allocholic acid, Tryptophan, Serotonin, Indolelactic acid
35	Byeon, et al., 2021 [97]	18 HC, 15 MCI, 17 AD	CSF	LC-MS	LPC, PC, LdMePE, LPE, dMePE, PE class
36	Liu, et al., 2021 [98]	19 HC, 25 AD	Brain tissue	LC-MS/MS	3-Methylguanine, Acetylcholine, Cytosine, Glycylproline, Guanosine, Histidinyl-aspartate, Imidazoleacetic acid, Indole-3-propionic acid, Inosine, Ketoleucine, Linoleamide, L-methionine, L-norleucine, L-valine, N6-methyladenosine, N-acetylglutamic acid, N-acetyl-L-aspartic acid, N-acetyl-L-phenylalanine, Palmitoleic acid, Phenylalanine, Phenylpyruvic acid, Piperidine, Sarcosine, Serylglycine, S-formylglutathione, Sphingosine, Theaflavin
37	Liu, et al., 2021 [99]	42 HC, 40 AD	Plasma	LC-MS/MS	Cer, ChE, DG, LPC, PC, PE, PI, SM, and TG class
38	Nielsen, et al., 2020 [100]	10 HC, 10 MCI, 10 AD	Plasma	LC-MS and NMR, spectroscopy	Valine, Histidine, Allopurinol riboside, Inosine, 4-Pyridoxic acid, Guanosine
39	Horgusluoglu, et al., 2022 [101]	362 HC, 270 early MCI, 494 late MCI, 298 AD	Plasma	HPLC-MS/MS	Short-chain Acylcarnitines/amino acids and medium/long-chain Acylcarnitines
40	Khan, et al., 2022 [102]	54 HC, 59 AD	Plasma	LC-MS/MS	PS (18:0/18:0), PS (18:0/20:0), PC (16:0/22:6), PC (18:0/22:6), PS (18:1/22:6)
41	Maffioli, et al., 2022 [103]	20 HC, 23 AD	Hippocampal tissue	LC-MS and GC-MS	Glycerol 3 phosphate, Erythrose 4 phosphate, Glucose, Deoxyuridine, Lactic acid, Saccharopine, Uridine 5′ diphosphate, Oxidized glutathione, Urea, Uracil, Succinic acid, Arginine, Beta D Glucose 6 phosphate, Glucosamine 6 phosphate, AICAR, Citrulline, Alanine, Pyruvic acid, Glutathione, Guanidoacetic acid, Mevalonate P, Lysine
42	Ozaki, et al., 2022 [104]	40 HC, 26 MCI, 40 AD	Plasma	CE-TOF-MS	Ornithine, Uracil, Lysine
43	Peña-Bautista, et al., 2022 [105]	20 HC, 11 AD, 31 MCI-AD	Plasma	UPLC-TOF/MS	Cer, LPE, LPC, MG, and SM were observed as being altered significantly between the preclinical AD and healthy groups. DG, MG, and PE were observed as being altered significantly between the MCI-AD and healthy groups.
44	Weng, et al., 2022 [106]	19 HC, 16 AD	Plasma	NMR spectroscopy	3-Phosphoglycerate, Fructose-6-phosphate, Glucose-6-phosphate, Betaine, Methyl-histidine, Glycerylphosphorylcholine, 2-Oxoglutarate, Citrate, Malate, Ergothioneine, Glutathione disulfide, Taurine, Carnosine, Ornithine, Glycine, Alanine, Serine, Glutamine, Tryptophan, Valine

**Table 2 ijms-24-04960-t002:** Summary of the included articles related to metabolomics studies performed on animal models.

	Reference	Study Population	Sample Type	Analytical Platform	Altered Metabolites
1	Salek, et al., 2010 [123]	Two age groups (2–3 months and 12–13 months) of transgenic CRND8 APP 695 and non-transgenic littermates (controls)	Brain samples with seven different brain regions	NMR-based metabolomics	Lactate, Aspartate, Glycine, Alanine, Leucine, Iso-leucine, Valine, N-acetyl-L-aspartate, Glutamate, Glutamine, Taurine, Gamma-amino butyric acid, Choline, Phosphocholine, Creatine, Phosphocreatine, Succinate
2	Hu, et al., 2012 [124]	50-week-old TASTPM transgenic AD mice and 5-month old wild type C57BL/6J mice	Plasma and brain tissues	GC-MS	AA, Androstenedione, Cortisol, D-fructose, D-galactose, D-glucose, Gluconic acid, Linoleic acid, L-serine, L-threonine, L-valine, Palmitic acid
3	Trushina, et al., 2012 [125]	APP/PS1 mice at 16 months of age and age-matched nTg controls	Hippocampus	GC-MS	Threonic acid, Ethanolamine, Alanine, Mannitol, Glycerol 3-P, Pyroglutamic acid, N-acetyl-aspartate, Lactic Acid
4	Tajima, et al., 2013 [126]	APP/tau mice at 4, 10, and 15 months of age and age-matched wild-type mice	Brain tissue, Plasma	RPLC-ESI-TOFMS	12-HETE, 19,20-diHDoPE, 17,18-diHETE, 19,20-EpDPE, 17,18-EpETE, Prostaglandin D2, 15-HETE, Phosphatidylcholines, PEs with Polyunsaturated fatty acids, Docosahexaenoyl (22:6) Cholesterol Ester (ChE), Ethanolamine Plasmalogens (pPEs), Sphingomyelins (SMs)
5	González-Domínguez, et al., 2014 [127]	6-month-old APP/PS1 mice and age-matched controls	Brain tissue with different regions	GC-MS and UPLC-MS	40 significant differences in metabolites related to abnormal purine metabolism, bioenergetic failures, dyshomeostasis of amino acids, and disturbances in membrane lipids
6	Kim, et al., 2014 [128]	Model: mice receive an intracerebroventricular infusion of Aβ, Control: age-matched untreated mice	Plasma	NMR-MS	Niacinamide, AMP, Hypoxanthine, Citrate, Lactate, Pyruvate, Creatine, Choline, Acetate, Phenylalanine, Glycine, Valine, Tyrosine, Alanine, Glucose, Glutamine
7	Lalande, et al., 2014 [129]	Single-transgenic Tg2576 mice at 1, 3, 6, and 11 months of age, and age-matched controls	Brain tissues with five different brain regions	NMR-MS	γ-Aminobutyric acid, Glutamate, N-acetylaspartate (NAA), Myo-Inositol, Creatine, Phosphocholine, Glu, Creatine, Taurine
8	González-Domínguez, et al., 2015 [130]	APP/PS1 mice at 6 months old, and age-matched nTg controls	Plasma	FI-APPI-MS, DI-ESI-MS	Choline, Serine, Valine, Threonine, Pyroglutamate, Creatine, Phosphoethanolamine, Histidine, Carnitine, Glucose, Tyrosine, Tryptophan, GPE, Inosine, Urea, Myristic Acid, Palmitoleic acid, HEPE, Lysophosphocholines, Phosphocholines, Diacylglycerols, Cholesteryl esters, Triacylglycerols
9	González-Domínguez, et al., 2015 [131]	APP/PS1 mice at 6 months of age and age-matched nTg controls	Plasma	GC-MS, UHPLC-MS	Phosphoethanolamine, Adenosine monophosphate, Citrulline, Citric acid, Monostearin, Bile acids, Eicosanoids, Fatty acid amides, Sphingoid bases, Lyso-phospholipids, Phospholipids, Sphingomyelins, Lactic acid, B-hydroxybutyric acid, Urea, Phosphoric acid, Glycine, Succinic acid, Threonine, Malic acid, Pyroglutamic Acid, Creatinine, Proline, Glycerol-3-phosphate, Citric acid, Glucose, Tyrosine, 6Myoinositol, Uric Acid, Glucose-6-phosphate, Tryptophan, Stearic acid, Myoinositol-1-phosphate, Serotonin, 1,3-Bisphosphoglycerate, Cholesterol
10	González-Domínguez, et al., 2015 [132]	6-month-old APP/PS1 mice and age-matched controls	Brain tissue with different regions	LC-QTOF-MS	52 significant differences in metabolites including phospholipids, fatty acids, purine and pyrimidine metabolites, acylcarnitines, sterols, and amino acids which related to the homeostasis of lipids, energy management, and metabolism of amino acids and nucleotides
11	Li, et al., 2015 [133]	Model: mice receive hippocampal region infusion of Aβ_1-42_, Control: age-matched untreated mice	Plasma, Brain tissues	UPLC-MS/MS	Phenylalanine, Tryptophan, Dihydrosphingosine, LPC (18:2), LPC (20:4), LPC (22:6), LPC (16:0), LPC (16:0), LPC (18:1), LPC (18:0)
12	Li, et al., 2016 [134]	APP/PS1 mice at 1 month old and age-matched nTg controls	Brain tissues	UPLC-MS	Hypoxanthine, Hexadecasphinganine, Dihydrosphingosine, Phytosphingosine, LPC (13:0), LPC (15:1), LPC (15:0), LPC (16:0), LPC (18:3), LPC (18:1), LPC (18:0)
13	Pan, et al., 2016 [135]	APP/PS1 mice at 1, 8, 10, 12, and 18 months of age, and age-matched nTg controls	Plasma and brain tissues	UPLC-TQ-MS	Phenylalanine, Trypotophan, Tyrosine, a-Aminoadipic acid, Asparagine, Histidine, Phenylalanine, Valine, Isoleucine, Methionine, Tyrosine, Methionine sulfoxide, Serotonin, Taurine
14	Nuriel, et al., 2017 [136]	APOE mice at 14.5 months of age and age-matched controls	Brain tissues	UPLC-MS	Myristic acid, DHA, Stearic acid, 12-Hydroxydodecanoic acid, Arachidic acid, Palmitic acid, Trisaccharide, Tetrasaccharide, Disaccharide, Phylloquinone, Tocopherol, Dehydroascorbic acid, Inosine 5′-monophosphate (IMP), D-fructose 6-phosphate, Succinoadenosine, Carnitine, Citric acid/Isocitric acid, Malic acid, ATP, Lanosterol, Cholesteryl 19acetate, Leucine, Proline, Glycine, Quinaldic acid, Kynurenine, Kynurenic acid, S-Adenosylhomocysteine, Carnosine, 4-Oxoproline, Tyramine, Thymidine, Uracil, GMP, Methylglutarylcarnitine, Trimethylamine N-oxide, 2-Hydroxypyridine, N-acetylneuraminic acid, Hydroxybutyric acid
15	Pan, et al., 2017 [137]	APP/PS1 mice at 6 and 12 months of age and age-matched controls	Plasma and brain tissues	LC-MS/MS	Cholic acid, Hyodeoxycholic acid, Lithocholic acid, Taurocholic acid, Tauromuricholic acid (α and β), Tauroursodeoxycholic acid, β-Muricholic acid, Ω-Muricholic acid
16	Bergin, et al., 2018 [138]	APP/PS1 mice at 7 and 13 months of age and age-matched nTg controls	Brain tissues, Plasma	HPLC-MS	Arginine, Citrulline, Ornithine, Agmatine, Putrescine, Spermine, Spermidine
17	Gao, et al., 2018 [139]	APP/PS1 mice at 12 months of age and age-matched nTg controls	Plasma	UPLC-MS	Acetoacetyl-CoA, 21-Deoxycortisol, 9 (10)-EpOME, 3α-Hydroxy-5β-androstan-17-one, PGF2α, Dehydroepiandrosterone, 3α,21-Dihydroxy-5β-pregnane-11,20-dione, Sphinganine, Sphingosine, DHA, Linoleic acid, 4,4-Dimethyl-5α-cholest-7-en-3β-ol, Arachidonic acid, PC (20:4 (5Z,8Z,11Z,14Z)/0:0), 9-Cis-retinoic acid, LTA4
18	Sun, et al., 2018 [140]	21-week-old APP/PS1 mice and age-matched controls	Plasma	UPLC-QTOF/MS	Arachidonic acid, Cer (d18:0/12:0), Glycerophosphocholine, Leukotriene B4, Linoleic acid, L-tryptophan, Phosphorylcholine, Phytosphingosine, Prostaglandin F2a, Prostaglandin G2, Sphinganine
19	Zheng, et al., 2018 [141]	APP/PS1 mice at 5 and 10 months of age and age-matched controls	Brain tissues	NMR-based metabolomics	Adenosine monophosphate, ADP, Alanine, Aspartate, Choline, Creatine/phosphocreatine, Glutamate, Glutamine, Glycerolphorylcholine, Glycine, Inosine, Inosine monophosphate, Lactate, Myo-inositol, N-acetylaspartate, Nicotinamide adenine dinucleotide, Phosphocholine, Succinate, γ-Aminobutyric acid
20	Zhou, et al., 2018 [142]	10-month-old APP/PS1 double transgenic mice and age-matched controls	Brain tissues	13C NMR based metabolomic	Glutamate, Glutamine, γ-Aminobutyric acid, Aspartate, Succinate, Lactate, Alanine
21	Chang, et al., 2019 [143]	5-month-old APP/PS1 mice and age-matched controls	Brain tissue with different regions	GC-TOF-MS	2-Hydroxyglutaric acid, 4-Guanidinobutyric acid, Aspartic acid, Beta-alanine, Citric acid, Creatinine, Ethanolamine, Fumaric acid, GABA, Glucose, Glucose-6-phosphate, Glutamic acid, Glycerol, Glycine, Inosine, Lactic acid, Leucine, Malic acid, N-acetyl-L-aspartic acid, Oleic acid, Proline, Pyroglutamic acid, Ribose-5-phosphate, Serine, Succinic acid, Threonine, Threonolactone, Tyrosine, Uracil, Urea, Valine
22	Li, et al., 2019 [144]	3xTg-AD mice at 19 months of age and age-matched nTg controls	Plasma	UPLC-Q/TOF-MS	Linoleic acid, 9 (10)-EpOME, DHA, Arachidonic acid, 11b-PGF2a, Sphingosine, Pyruvate, L-Leucine, Ursocholic acid, Corticosterone, L-Tryptophan, Dodecanoic acid, L-lysine
23	Liu, et al., 2019 [145]	APP/PS1 mice at 6 and 9 months of age and age-matched controls	Pancreas and serum	NMR-based metabolomics	2-Aminobutyrate, Acetate, Adenosine-monophosphate, AMP, Adenosine-triphosphate, ATP, Alanine, Allantoin, Aspartate, Betaine, Choline, Citrate, Fumarate, Glucose, Glucose 1-phosphate, Glutamine, Glycine, Lactate, Leucine, Malate, Nicotinurate, Pyruvate, Reduced nicotinamide adenine dinucleotide phosphate, NADPH, Serine, Succinate, Taurine, Valine
24	Liu, et al., 2019 [146]	3-month-old C57BL/6 mouse with local administration of Aβ peptide into brain, and 3-month-old controls	Hippocampus	UPLC-QTOF/MS	Adenine, Adenosine, Citrulline, Inosine, L-Arginine, L-Isoleucine, L-Threonine, Normetanephrine
25	Pan, et al., 2019 [147]	8-month-old PLB4 mice and age-matched controls	Brain tissues	LC-MS	Leucine, Creatinine, Putrescine and species of Acylcarnitines, LysoPC, PCs, Sphingomyelin
26	Rong, et al., 2019 [148]	Model: SD rats (200 ± 20 g) induced by D-Gal and Aβ_25–35_ injection, Control: age-matched untreated rats	Plasma, Brain tissue	UPLC-Q/TOF-MS	LysoPC (17:0), LysoPC (20:2), LysoPC (20:4), LysoPC (16:0), LysoPC (18:0), LysoPE (20:4), PE (18:1), PE (22:4)
27	Hunsberger, et al., 2020 [149]	APP/PS1 mice at 6, 12, and 24 months of age and age-matched controls	Prefrontal cortex, Hippocampus, Spleen	LC-MS and GC-MS	1-Methyl nicotinamide, 2-Methylbutyroylcarnitine, 7-Methylguanine, Arginine, Argininosuccinic acid, Azelaic acid, Dimethylglycine, Formiminoglutamic acid, Glutathione, Glycerate, Glycolic acid, Guanidinobutanoic acid, Hydroxyproline, Hydroxypyruvic acid, Isovalerylcarnitine, L-glutamic acid, L-methionine, L-tyrosine, Methylhistamine, Methylhistidine, N6-acetyl-L-lysine, N-acetyl L-aspartic acid, N-acetylneuraminic acid, Orotate, Phenylpyruvic acid, Phosphocreatine, Propionylcarnitine, Pyroglutamic acid, Pyruvate, S-adenosylmethionine, SAICAR, Sphingosine-1-phosphate, Uracil, Uridine
28	Kim, et al., 2020 [150]	5xFAD mice at 8 and 12 months of age and age-matched controls	Hippocampus	UPLC-MS	Nicotinamide, Adenosine monophosphate, LysoPC (16:0), LysoPC (18:0), and LysoPE (16:0)
29	Sun, et al., 2020 [151]	Model: SD rats (weight 260 ± 20 g) receive an infusion of Aβ_1-42_. Control: age-matched untreated rats.	Plasma	HPLC-MS	Arachidic acid, FA (18:0), FA (18:1), FA (18:2), FA (18:3), CE (20:5), CE (22:6), Cer (d18:0/20:0), Cer (d18:1/24:0), MG (16:0/0:0/0:0), DG (18:0/16:0/0:0), TG (16:0/18:0/18:1), LPC (18:0), LPC (20:4), LPC (20:5), LPC (20:1), LPC (O-18:0), LPC (P-16:0), LPC (P-18:0), PC (18:2/18:0), PC (18:2/18:2), PC (18:2/20:4), PC (35:1), PC (35:4), PC (36:1), PC (O-32:0), SM (d18:0/16:1), SM (d40:2)
30	Tondo, et al., 2020 [152]	4-month-old 16 rTg4510 mice and age-matched controls	Brain	LC-MS	Glutamine, Serotonin, Sphingomyelin C18:0
31	Yi, et al., 2020 [153]	SD rats (180 ± 40 g) were induced by Aβ_1-42_ injection, Control: age-matched untreated rats	Prefrontal cortex	UPLC-TripleTOF/MS	Cer (d18:1/16:0), Cer (d18:1/24:1 (15Z)), GlcCer (d14:1/20:0), Galbeta-cer (d18:1/20:0), Cer (d18:1/18:0), PI (18:0/0:0), LysoPE (16:1 (9Z)/0:0), Glucosylceramide (d18:1/18:0), PE (18:1 (9Z)/16:0), LysoPC (16:1 (9Z)/0:0), PI (20:4 (5Z,8Z,11Z,14Z)/0:0), PE (16:1 (9Z)/P-16:0), 4-Nitrophenol, LysoPE (0:0/22:6 (4Z,7Z,10Z,13Z,16Z,19Z)), PE (15:0/14:1 (9Z)), Cer (d18:1/20:0), PC (18:1 (11Z)/18:2 (9Z,12Z)), PS (20:1 (11Z)/18:0), PC (18:1 (11Z)/20:4 (5Z,8Z,11Z,14Z)), PE (18:3 (9Z,12Z,15Z)/22:1 (13Z)), Cer (d18:0/22:0), PE (15:0/22:0)
32	Zhang, et al., 2020 [154]	APP/tau mice at 7 months of age and age-matched wild-type mice	Brain tissue	UHPLC-QTOF/MS	Hydroxy-E4-neuroprostane, Prostaglandin D3, PS (18:0/22:5), PC (14:0/16:0), PC (16:0/19:0), PE (22:5/22:6), PE (15:0/22:4), PE (18:1/20:2), Leukotriene D4, Cer (d18:1/18:1)
33	Zhang, et al., 2020 [155]	APP/tau mice at 2, 3, and 7 months of age and age-matched wild-type mice	Brain tissue, Plasma	UHPLC-QTOF/MS	Lysophospholipids, PCs, Pes, Cer, Fatty acids, Diacylglycerols (DGs), Triacylglycerols (TGs)
34	Dejakaisaya, et al., 2021 [156]	Six-month-old mixed-sex Tg2576 mice and c57 × SJL (WT) littermates	Cortex	LC-MS	2′-Deoxyguanosine 5′-monophosphate, Lysophosphatidylethanolamines (18:0), Diglyceride (P-32:1), 3–4-Hydroxyphenyllactate, Npi-methyl-l-histidine, O-butanoylcarnitine, O-propanoylcarnitine, Phosphatidylethanolamine (44:3), Phosphatidylserine (36:2), Phosphatidylserine (40:7), Phosphatidylserine (44:12)
35	Speers, et al., 2021 [157]	8-month-old 5xFAD mice and age-matched controls	Cortical tissue	HPLC-MS	Betaine, Lysine, Pyridoxamine, Urate, NAD, Erythritol, Spermine, N-Acetylmannosamine, Glycerol 2-phosphate, Lauroyl-L-carnitine, Nicotinamide hypoxanthine Dinucleotide, Sodium taurocholate, Homocysteine, Betaine, N-acetyl-mannosamine, S-adenosyl-homocysteine, Adenosine 3′5′-cyclic monophosphate
36	Sun, et al., 2021 [158]	Model: SD rats with Aβ_1-42_ protein injection. Control: age-matched untreated rats.	Brain tissue	HPLC/FT-ICR MS	DG (20:4/16:0), DG (22:6/16:0), TG (10:0/18:0/16:0), TG (16:0/16:0/18:1), TG (43:1), TG (16:0/14:0/16:1), Glucosylceramide (d18:1/24:1), PA (22:6/22:5), PG (18:0/22:4), PG (40:5), PG (43:6), PI (18:0/20:4), PI (36:4), PI (18:1/20:4), PS (18:0/20:4), LPE (22:6), PC (17:1/18:1), PC (20:0), PC (34:5), Adrenic acid
37	Zhao, et al., 2021 [159]	3×Tg-AD mice at 2 and 6 months of age and age-matched nTg controls	Hippocampus	UPLC-MRM-MS	84 significant differences in metabolites related to abnormal purine, pyrimidine, arginine, and proline metabolism, glycerophospholipid metabolism
38	Cheng, et al., 2022 [160]	9-month-old male APP/PS1 mice and age-matched controls	Plasma	UHPLC-MS/MS	3-Indole carboxylic acid glucuronide, Cyclic dAMP, (5Z,9Z)-2-Methoxy-hexacosadienoic acid, Caffeoyl aspartic acid, Saccharin, (Z)-13-Oxo-9-octadecenoic acid, 5C-Aglycone, PS (O-16:0/20:2 (11Z,14Z)), Bisacurone Epoxide, PS (22:0/15:0), 6-Hydroxymelatonin glucuronide, Methyl 2-undecynoate, Cis-3-hexenyl trans-4-hexenoate
39	Dai, et al., 2022 [161]	4-month old APPswe/PS1∆E9 (PAP) transgenic mice and age-matched controls	Brain and plasma	LC-MS	23-Nordeoxycholic acid, 23-Norcholic acid, 7-Ketodeoxycholic acid, Deoxycholic acid, β-muricholic acid, Cholic acid, Cholecalciferol, Pyridoxal, Flavin adenine dinucleotide, Riboflavin, Arachidic acid, Eicosapentaenoic acid, Neuronic acid, Erucic acid, Acetoacetate, Fumaric acid, Estradiol, Progesterone, Testosterone
40	Dehghan, et al., 2022 [162]	50-week-old Abca7 knockout mice and age-matched controls	Brain	UPLC-MS	LacCer, Sphingomyelins, Cers, Hexosylceramides
41	Dunham, et al., 2022 [163]	4, 8, 12, and 18 months 5xFAD and wild-type littermates	Plasma	UPLC-MRMs	Glycine, Carnosine, Serine, SM C24:1, Serotonin, Serotonin, Spermidine, Spermine, Aspartic acid phosphatidylcholines (PCs), α-Aminoadipic acid, Serotonin, Glutamic acid
42	Sun, et al., 2022 [164]	4-month-old APP/PS1 mice and age-matched controls	Serum, Brain Tissues, Feces	LC-MS	88 significant differences in metabolites related to neurotransmitters metabolism, lipid metabolism, aromatic amino acids metabolism, energy metabolism, vitamin digestion and absorption, and bile metabolism

## Data Availability

Not applicable.

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
