# Peer review of "Status of Metabolomic Measurement for Insights in Alzheimer’s Disease Progression—What Is Missing?"

_ijms, 2023, doi:10.3390/ijms24054960_

Round 1
Reviewer 1 Report
In this review, the authors present an extensive amount of data, focusing on metabolomic studies by integrating and analyzing the reported results. It is well-organized and well written. However, some aspects need to be clarified before it can be further considered for publication.
1. The section "2.2.1 Metabolomics in AD Human Studies" lacks a concluding paragraph summarizing or highlighting key findings. Table 1 is informative, but an integrative conclusion is needed for this section.
2. The same applies to the section "2.2.2 Metabolomics in Animal Models of AD".
3. In addition, it would be of great interest to design a figure with two panels comparing the major altered metabolites in AD between human studies and animal models.
4. An important aspect is the nature of the change in metabolic pathways: are they up-regulated or down-regulated? Are common metabolic pathways enhanced or diminished? In the review, this aspect is mentioned only in lines 265-270. It would be useful to collect these data from all the articles analyzed.
5. The 8 metabolic pathways mentioned in section 4 that are consistent between AD subjects and AD mouse models, are they from measurements in CFS or tissue or plasma? Because later it is mentioned that 12 common metabolic pathways were altered in plasma. Please clarify.
6. If 8 or 12 common metabolic pathways were altered between AD subjects and AD mouse models, why were a total of nine described from 4.1 to 4.9 (not eight or twelve )?
Author Response
1.The section "2.2.1 Metabolomics in AD Human Studies" lacks a concluding paragraph summarizing or highlighting key findings. Table 1 is informative, but an integrative conclusion is needed for this section.
Reply: Thank you for your comment. A conclusion paragraph has been added to this section.
2.The same applies to the section "2.2.2 Metabolomics in Animal Models of AD".
Reply: Thank you for your comment. A conclusion paragraph has been added to this section.
3.In addition, it would be of great interest to design a figure with two panels comparing the major altered metabolites in AD between human studies and animal models.
Reply: I appreciate your comment and understand your meaning. However, the coverage of metabolites varied among different studies due to different analytical platforms having been used (e.g., NMR, GC-MS, LC-MS) and thus the analytical methods differ in different laboratories. The different metabolites between human studies and animal studies may be due to the varied detection sensitivity.
4.An important aspect is the nature of the change in metabolic pathways: are they up-regulated or down-regulated? Are common metabolic pathways enhanced or diminished? In the review, this aspect is mentioned only in lines 265-270. It would be useful to collect these data from all the articles analyzed.
Reply: We thank the reviewer for this important comment. Indeed, within pathways the measured metabolites may be up- or down-regulated. However, we could not say this pathway was up- or down-regulated overall, as we can only make a statement on the measured metabolites themselves. Further, certain metabolites may vary between different matrices in one study, more research needs to be done to have a conclusive answer on this.
5.The 8 metabolic pathways mentioned in section 4 that are consistent between AD subjects and AD mouse models, are they from measurements in CFS or tissue or plasma? Because later it is mentioned that 12 common metabolic pathways were altered in plasma. Please clarify.
Reply: Thank you for your comment. Because 3.1 and 3.3 are using all matrices, the 8 metabolic pathways in common between AD subjects and AD mouse models mentioned in section 4 are the analysis result by using all matrices (brain, plasma, and CSF). I adjusted the description in section 4.
6.If 8 or 12 common metabolic pathways were altered between AD subjects and AD mouse models, why were a total of nine described from 4.1 to 4.9 (not eight or twelve)?
Reply: Thank you for your comment. Since some pathways can be combined and described together, the described pathways in the discussion are not equal to 8 or 12. In the discussion section we describe a total of 10 pathways with relevant biology, including i) arginine metabolism, ii) alanine, aspartate, and glutamate metabolism, iii) purine metabolism, iv) taurine and hypotaurine metabolism, v) Cholinergic System, vi) Fatty acids, vii) Glycerolipids, viii) Glycerophospholipids, ix) Sphingolipids, x) Cholesterol and cholesteryl esters.

Reviewer 2 Report
In the manuscript entitled “Status of metabolomic measurement for insights in Alzheimer’s disease progression - what is missing?”, Yin et al. summarized and analyzed the already available metabolomics data obtained from different types of biological samples of AD patients and animal models of AD. The manuscript is comprehensive, very well-written and structured and it really provides the current status of metabolomics applications in studying Alzheimer's disease.
Author Response
Dear reviewer, thank you very much for your kind comments and the time taken to evaluate our manuscript.

Reviewer 3 Report
Review presents comprehensive encyclopedia-type collection of large number of metabolomic studies on MCI, AD humans and transgenic animals. It is valuable information itsef. However, the interpretation of these extensive data in terms of pathomechanisms is limited what is natural in accord with presentation scheme. Glutamatergic system contribution to AD is described in details. However, that does not allow to look into compartmentalization of degeneration processes between neuronal and different classes of glial cells and on subcellular level as well. Cholinergic system disturbances has been omitted , what should be presented if one wants to look into cognitive deficits being a hallmark of AD and MCI.
Lines 202-203. CSF does not supply nutrients into the brain cells. It takes place in the interstitial space supplied by transporters on BBB and cell plasma membranes. This pitfall has to be corrected.
Review, although being OK as whole needs some adjustment to known pathomechanisms linked with metabolism of particular metabolites.
Author Response
Thank you for your time reviewing and evaluating our manuscript and your helpful comments.
1.Glutamatergic system contribution to AD is described in details. However, that does not allow to look into compartmentalization of degeneration processes between neuronal and different classes of glial cells and on subcellular level as well.
Reply: We thank the reviewer for this remark. Indeed, as the reviewer mentions, glutamate is considered as the main neurotransmitter of neocortical and hippocampal pyramidal neurons and is involved in higher mental functions such as cognition and memory. In this review, we investigated plasma, brain and other tissues but did not focus on the cellular level or the interaction between different cell types.
2.Cholinergic system disturbances has been omitted, what should be presented if one wants to look into cognitive deficits being a hallmark of AD and MCI.
Reply: Thank you for your comment. We agree that cholinergic system has been omitted. One paragraph of the description of cholinergic system was added in the discussion part to address this.
3.Lines 202-203. CSF does not supply nutrients into the brain cells. It takes place in the interstitial space supplied by transporters on BBB and cell plasma membranes. This pitfall has to be corrected.
Reply: Thank you. We agree with your comment and the sentence (CSF can transport nutrients to maintain homeostasis of neuronal cells and remove waste products of neuronal activity from the brain) has been removed.
4.Review, although being OK as whole needs some adjustment to known pathomechanisms linked with metabolism of particular metabolites.
Reply: Thank you for your time and valuable comments, which have helped us improve the manuscript. We hope that the reviewer agrees with us.
